# Increasing β-catenin/Wnt3A activity levels drive mechanical strain-induced cell cycle progression through mitosis

Blair W Benham-Pyle[1], Joo Yong Sim[2†], Kevin C Hart[3], Beth L Pruitt[2,4,5], William James Nelson[1,3,5*]

[1]Program in Cancer Biology, Stanford University, Stanford, United States; [2]Department of Mechanical Engineering, Stanford University, Stanford, United States; [3]Department of Biology, Stanford University, Stanford, United States; [4]Stanford Cardiovascular Institute, Stanford University, Stanford, United States; [5]Department of Molecular and Cellular Physiology, Stanford University, Stanford, United States

**Abstract** Mechanical force and Wnt signaling activate β-catenin-mediated transcription to promote proliferation and tissue expansion. However, it is unknown whether mechanical force and Wnt signaling act independently or synergize to activate β-catenin signaling and cell division. We show that mechanical strain induced Src-dependent phosphorylation of Y654 β-catenin and increased β-catenin-mediated transcription in mammalian MDCK epithelial cells. Under these conditions, cells accumulated in S/G2 (independent of DNA damage) but did not divide. Activating β-catenin through Casein Kinase I inhibition or Wnt3A addition increased β-catenin-mediated transcription and strain-induced accumulation of cells in S/G2. Significantly, only the combination of mechanical strain and Wnt/β-catenin activation triggered cells in S/G2 to divide. These results indicate that strain-induced Src phosphorylation of β-catenin and Wnt-dependent β-catenin stabilization synergize to increase β-catenin-mediated transcription to levels required for mitosis. Thus, local Wnt signaling may fine-tune the effects of global mechanical strain to restrict cell divisions during tissue development and homeostasis.

*For correspondence: wjnelson@stanford.edu

Present address: †Electronics and Telecommunications Research Institute, Daejeon, Republic of Korea

## Introduction

Mechanical cues are critical for regulating cellular growth, morphology, and behavior in developing and adult tissues. Recently, cadherin-mediated cell-cell adhesions have been identified as core components that regulate the response of multicellular tissues to externally applied strain (*Whitehead et al., 2008*; *Desprat et al., 2008*; *Kim et al., 2011*). Significantly, the cadherin-associated transcriptional co-activator β-catenin accumulates in the nucleus following mechanical strain (*Farge, 2003*; *Benham-Pyle et al., 2015*).

β-Catenin transcriptional activity drives cellular proliferation and is a downstream effector of the Wnt signaling pathway (*Nelson and Nusse, 2004*; *Clevers and Nusse, 2012*). β-Catenin protein levels in cells are tightly controlled by sequestering β-catenin at cell-cell contacts and active degradation of cytosolic β-catenin (*Whitehead et al., 2008*; *Desprat et al., 2008*; *Kim et al., 2011*; *Aberle et al., 1997*; *Ikeda et al., 1998*; *Su et al., 2008*). The E-cadherin-bound pool of β-catenin is regulated by a balance of tyrosine kinase and tyrosine phosphatase activities (*Farge, 2003*; *Benham-Pyle et al., 2015*; *Muller et al., 1999*; *Piedra et al., 2001*; *Lilien and Balsamo, 2005*; *Tan et al., 2016*). The affinity between E-cadherin and β-catenin at junctions is decreased by the activity of cytoplasmic and receptor tyrosine kinases (EGFR, Src, Abl, Fyn/Fer) (*Nelson and Nusse,*

**eLife digest** Tissues and organs can both produce and respond to physical forces. For example, the lungs expand and contract; the heart pumps blood; and bones and muscles grow or shrink depending on how much they are used. These responses are possible because cells contain proteins that can respond to physical forces. One of the best studied of these is a protein called β-catenin, which increases the activity of genes that trigger cells to divide to promote the expansion of tissues. β-catenin is over-active in many types of cancer cells where it contributes to tumor growth. In addition to being switched on by mechanical force, β-catenin is also activated when cells detect a signal molecule called Wnt.

Cells cycle through a series of stages known as the cell cycle to ensure that they only divide when they are fully prepared to do so. Benham-Pyle et al. investigated if physical force and Wnt activate β-catenin in the same way or if they have different effects on cell division. The experiments were conducted on dog kidney cells that had left the cell cycle and had therefore temporarily stopped dividing. Physical forces, such as stretching, resulted in β-catenin being modified by an enzyme called SRC kinase, which allowed the cells to re-enter the cell cycle. On the other hand, Wnt stabilized β-catenin and temporarily increased the number of cell divisions.

When mechanical stretch and Wnt signaling were combined, the cells were more likely to re-enter the cell cycle and divide compared to either stimulus alone. These data suggest that physical force and Wnt signaling affect β-catenin differently and that they can therefore have a greater effect on cell or tissue growth when they act together than on their own.

The findings of Benham-Pyle et al. show that β-catenin is not simply switched on or off, but can have different levels of activity depending on the input the cells are receiving. Future experiments will test whether these mechanisms also exist in three-dimensional tissues, which will help us understand how organs develop.

*2004*; *Clevers and Nusse, 2012*; *Matsuyoshi et al., 1992*; *Takeda et al., 1995*; *Coluccia et al., 2007*; *Krejci et al., 2012*) and increased by the activity of tyrosine phosphatases (PTB1B, SHP-3, PTP, LAR) (*Lilien and Balsamo, 2005*; *Tan et al., 2016*; *Sap et al., 1994*; *Hellberg et al., 2002*; *Xu et al., 2002*). In normal epithelia, cytoplasmic levels of β-catenin are very low due to targeting of β-catenin for degradation through β-catenin phosphorylation on Serine 45 by Casein Kinase I (CKI), which primes β-catenin for the further phosphorylation of Serine 33, Serine 37, and Threonine 41 by Glycogen Synthase Kinase 3β (GSK3β) (*Aberle et al., 1997*; *Amit et al., 2002*; *Liu et al., 2002*). CKI/GSK3β phosphorylated β-catenin is ubiquitinated within the Axin/Adenomatous Polyposis coli complex, and then degraded in the proteasome (*Winston et al., 1999*).

Accumulation of cytosolic β-catenin can be triggered by Wnt, the mammalian homolog of *Drosophila Wingless*. When Wnt ligand binds to Frizzled/LRP receptors, Axin is recruited to the plasma membrane and CKI/GSK3β are inactivated, resulting in inhibition of β-catenin ubiquitination and degradation (*Li et al., 2012*; *Vinyoles et al., 2014*). As a result, β-catenin accumulates in the cytoplasm, translocates to the nucleus, and activates target gene transcription in a complex with Tcf/Lef (*Clevers and Nusse, 2012*). Wnt/β-catenin activation broadly promotes cell proliferation and tissue expansion during developmental patterning. Consequently, mis-regulation of the Wnt signaling pathway, particularly β-catenin degradation, and downstream β-catenin transcriptional activity disrupt contact-mediated cell cycle inhibition, drive epithelial to mesenchymal transition, and promote cellular transformation and cancer progression (*Orford et al., 1999*; *Korinek et al., 1997*; *Clevers, 2006*; *Chen et al., 2012*).

In addition to its known roles in cadherin-mediated cell-cell adhesion and as a downstream effector of Wnt-mediated proliferation, β-catenin is a mechanotransducer. Mechanical strain induces nuclear accumulation and increased transcriptional activity of β-catenin (*Benham-Pyle et al., 2015*; *Sen et al., 2008*), and mechanical perturbation of living tissues is associated with increased β-catenin signaling and induction of downstream transcriptional targets (*Farge, 2003*; *Farge and development, 2011*). β-Catenin transcriptional activation downstream of mechanical force or Wnt stimulation

has been studied separately. It remains unknown whether β-catenin's role as a target of mechanical force is independent, redundant or synergistic to Wnt signaling in tissue homeostasis.

Previously, we showed that mechanical strain drives the re-entry of a quiescent epithelial monolayer into the cell cycle by sequential, but independent induction of Yap1 (G0 to G1) and then β-catenin transcriptional activities (G1 to S): inhibition of Yap1 and β-catenin transcriptional activities was sufficient to block cell cycle entry and G1 to S transition, respectively (*Benham-Pyle et al., 2015*). However, cells that exit quiescence following mechanical strain accumulate in S/G2 and do not enter mitosis, suggesting that a further activation event is required to complete cell cycle progression. Here we show that mechanical strain-induced Src phosphorylation of β-catenin and Wnt3A pathway-dependent stabilization of cytoplasmic β-catenin synergize to increase β-catenin transcriptional activity to levels that drive cell cycle progression through S/G2 and mitosis.

## Results

### Live imaging of epithelial monolayers reveals that mechanical strain drives cell-cycle re-entry, but does not increase cell divisions

Physical constraints and mechanical cues regulate cell proliferation in multicellular tissues. In cell culture models of quiescent or growing epithelial cell monolayers, mechanical strain induces cell cycle re-entry and increases the number of actively cycling cells (*Benham-Pyle et al., 2015*; *Streichan et al., 2014*). To test directly if mechanical strain is sufficient to drive cells into mitosis, super-confluent monolayers of quiescent normal kidney epithelial (MDCK) cells (*Benham-Pyle et al., 2015*) were formed on flexible silicone substrates either in a single-well biaxial cell stretching device compatible with live imaging (*Figure 1—figure supplement 1*), or in an integrated strain array (ISA) (*34*, see also Materials and methods). The design and fabrication of the biaxial cell stretching device compatible with live imaging allowed for direct visualization of strained monolayers with an inverted fluorescence microscope. Briefly, quiescent monolayers were formed on compliant silicone substrates in a PDMS well, surrounded by a pneumatic chamber separated by a thin silicone wall. Vacuum pressure applied to the pneumatic chamber deflected the silicone wall outwards, resulting in biaxial stretch accompanied by equi-biaxial in-plane strain (for details, see *Figure 1—figure supplement 1*, and Materials and methods). The live cell stretcher and ISA were able to apply maximum strains of 8.5 and 15%, respectively. The maximum level of static biaxial stretch was applied and held for up to 24 hr, and cells were either imaged live or fixed (ISA), and then processed for quantitative image analysis using MATLAB scripts which enabled unbiased image quantitation of large numbers of cells (see Materials and methods).

A fluorescence ubiquitination-based cell cycle indicator (Fucci MDCK-2, see [*Streichan et al., 2014*]) was used to monitor cell cycle dynamics following mechanical strain. Fucci MDCK cells stably express mKO2-Cdt1 (red fluorescence) during G0 and G1 phases, and mAG-Geminin (green fluorescence) beginning at S and ending at mitosis when Geminin is degraded. Thus, the level of mAG-Geminin fluorescence indicates time from entering into S, and loss of mAG-Geminin fluorescence marks entry into mitosis; the transition in cell fluorescence over time from red to green to red marks the transition of cells from G1 into S, then S/G2 into mitosis, and the re-entry of daughter cells into G1, respectively.

In the absence of mechanical strain, quiescent epithelial monolayers maintained a steady turnover rate over 24 hr that was characterized by a low, but constant number of cells in S/G2 (~10% Geminin-positive, *Figure 1A,C*, *Video 1*) and mitosis (~1 division/hour/0.1 mm$^2$, *Figure 1B,D*, *Video 1*). Upon application of mechanical strain, there was an immediate, small, but statistically significant increase in the number of Geminin-positive cells (*Figure 1A,C*, see also [*Streichan et al., 2014*], *Video 2*) that did not increase further until 8 hr following strain when there was a constant, linear increase through 24 hr; however, there was not a significant increase in the number of cells entering mitosis (*Figure 1C*, *Video 2*). A previous study reported that the fraction of mitotic cells in a suspended MDCK cell monolayer was also very low (~0.5%) and increased slightly (~2.5%) upon significantly higher levels of strain (~30%) than used here (*Wyatt et al., 2015*). Since mechanical strain-induced cell cycle re-entry results in cells entering S phase 6–8 hr following application of strain (*Benham-Pyle et al., 2015*; *Aragona et al., 2013*), an increase in Geminin-positive cells at 8 hr is

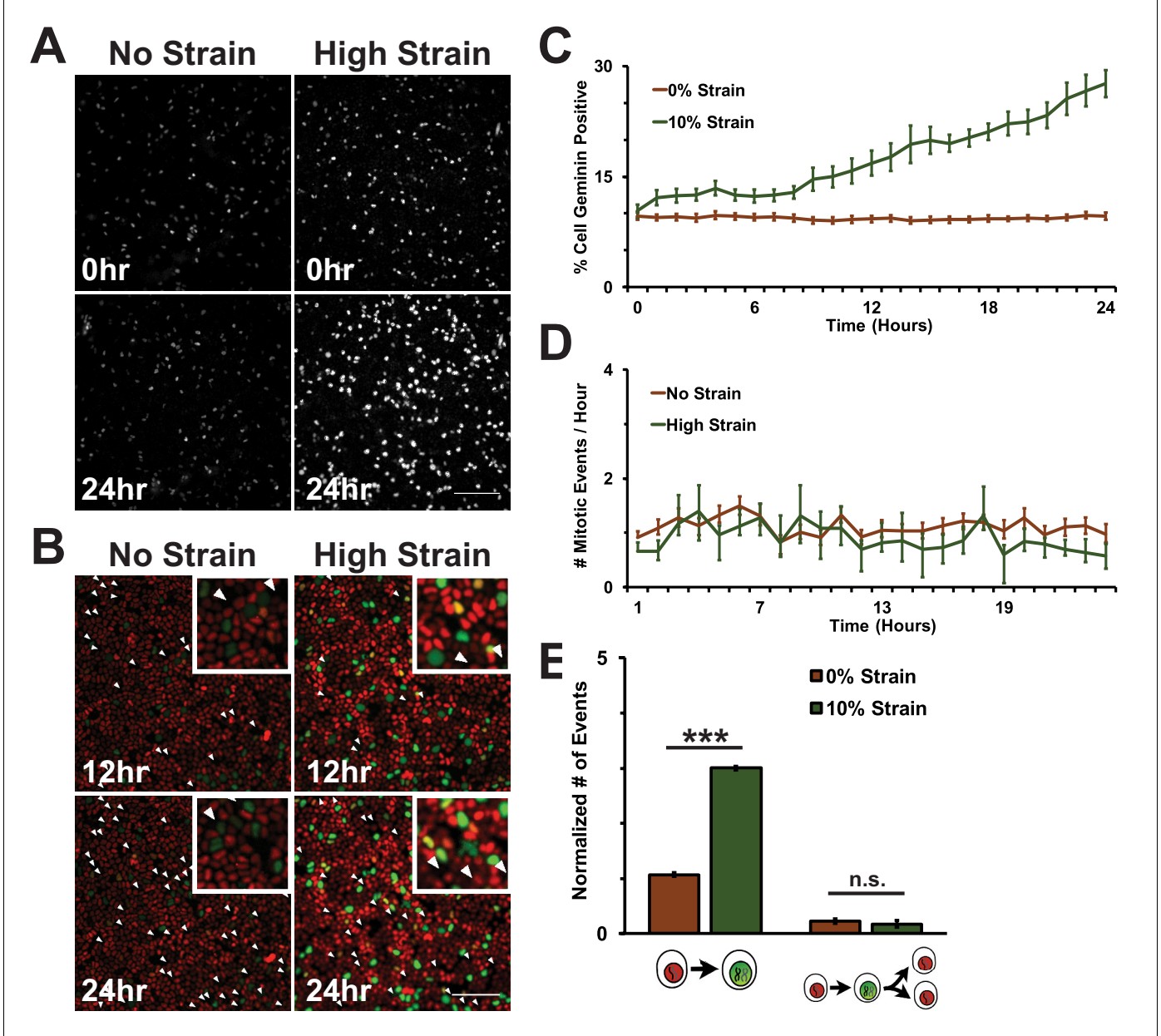

**Figure 1.** Mechanical strain is sufficient to drive cell cycle re-entry, but not entry into mitosis. (**A**) Distribution of mAG-Geminin in Fucci MDCK monolayers 0 hr or 24 hr after the application of No Strain or High Strain (~8.5%) using the biaxial live cell stretcher. Scale bar: 150 µm. (**B**) Distribution of mitotic events (white arrow heads) in Fucci MDCK monolayers 12 hr or 24 hr after the application of No Strain or High Strain (~8.5% Strain) using the biaxial live cell stretcher. Scale bar: 100 µm. (**C**) Quantification of percent Geminin positive cells in Fucci-MDCK monolayers; high strain is statistically significant (p<0.05) relative to no strain from 1–24 hr. (**D**) Number of mitotic events per hour in Fucci-MDCK monolayers; there is no statistically significant difference at any time point. (**E**) Single cell tracking quantification of the number of cell objects accumulated in S/G2 (red to green fluorescence, left) or passed through S/G2 and divided (red to green to red fluorescence, right) during 24 hr. All quantifications were from at least 3 independent experiments and included analysis of at least 9500 cells. Quantifications were mean +/- SEM; unpaired t-test p values<0.001 (***).

The following source data and figure supplements are available for figure 1:

**Source data 1.** Data used to construct graphs in *Figure 1* and *Figure 1—figure supplement 1*.

**Figure supplement 1.** Design and calibration of bi-axial live cell stretcher.

**Figure supplement 2.** Mitotic events and distance to nearest neighbor in fixed MDCK monolayers after mechanical strain.

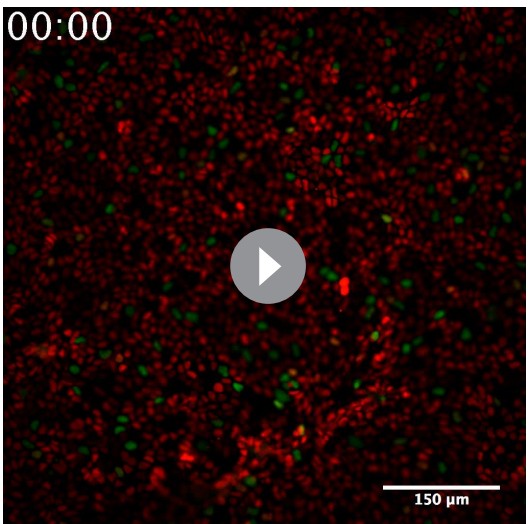
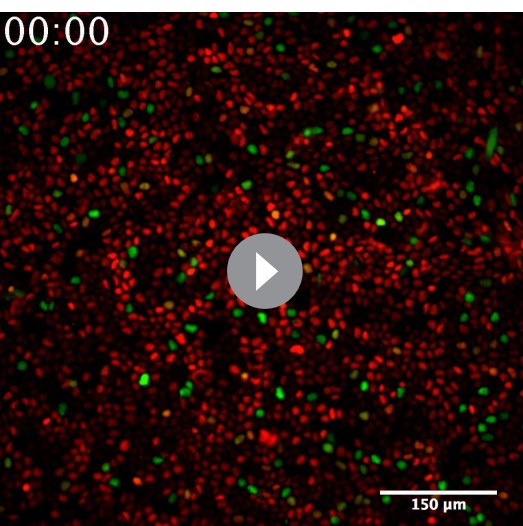

**Video 1.** Representative movie of a Fucci-MDCK monolayer treated with DMSO after no strain. Same data as in *Figures 1*, *5*, and *6*.

**Video 2.** Representative movie of a Fucci-MDCK monolayer treated with DMSO after mechanical strain. Same data as in *Figures 1*, *5*, and *6*.

consistent with an increase in the number of cells that had exited quiescence (G0), proceeded through G1, and then entered S.

Despite an increase in cells progressing through G1 and S upon mechanical strain, the number of observed mitotic events remained similar to unstrained monolayers (*Figure 1B,D*, *Video 1* and *Video 2*). This was confirmed by single cell tracking of Geminin-positive cells: the number of cells that transitioned from G1 to S (red to green fluorescence) increased with the application of mechanical strain (*Figure 1E*, left), but the number of cells that transitioned from G1 to S and then divided (red to green to red fluorescence, *Figure 1B*, white arrows) did not change (*Figure 1E*, right). Imaging of fixed MDCK monolayers using the ISA also revealed no difference in the number of mitotic events observed between monolayers that were mechanically strained or not (*Figure 1—figure supplement 2*). Together, these results indicate that mechanical strain of quiescent epithelial monolayers resulted in cell cycle re-entry and the accumulation of cells in S/G2 but, significantly, these cells did not enter mitosis.

The prolonged S/G2 phase in cells following mechanical strain could be due to the accumulation of DNA damage and activation of the DNA damage checkpoint. This cell cycle checkpoint regulates cell-cycle arrest by activating DNA repair pathways, inducing cell death by apoptosis (*Zhou and Elledge, 2000*) and inhibiting mitotic entry (*Furnari et al., 1997*; *Liu et al., 2000*). The small surface area (0.81 cm$^2$) of the ISA does not provide sufficient cell numbers for biochemical characterization (see Materials and methods). Therefore, to test whether mechanical strain induced an accumulation of cells in S/G2 due to strain-induced DNA damage, monolayers were stained for the DNA damage associated histone variant phospho-γH2A.X (*Rogakou et al., 1998*; *McManus and Hendzel, 2005*), and the DNA repair proteins p53 and p53 binding protein 1 (p53BP1) (*Zhang et al., 2009*; *Wagstaff et al., 2016*). Unstrained, quiescent MDCK monolayers had very low levels of phospho-γH2A.X, p53, and p53BP1 (*Figure 2A,C E*). Upon mechanical strain, the level of DNA damage – indicated by increased γH2A.X staining – increased marginally (*Figure 2A,C*), but levels of p53 and p53BP1 did not (*Figure 2C–F*). The level of DNA damage (γH2A.X positive cells) in mechanically strained monolayers was considerably less than monolayers treated with the DNA damage inducing agent MMS (*Figure 2A,B*) or in actively cycling cells (*Figure 2A,B*). Similarly, p53 and p53BP1 levels were significantly higher in MMS treated monolayers (*Figure 2C–F*) and in actively cycling cells, compared to strained and un-strained quiescent monolayers. Thus, the increase in DNA damage observed with mechanical strain was likely the result of increased DNA synthesis upon cell cycle re-

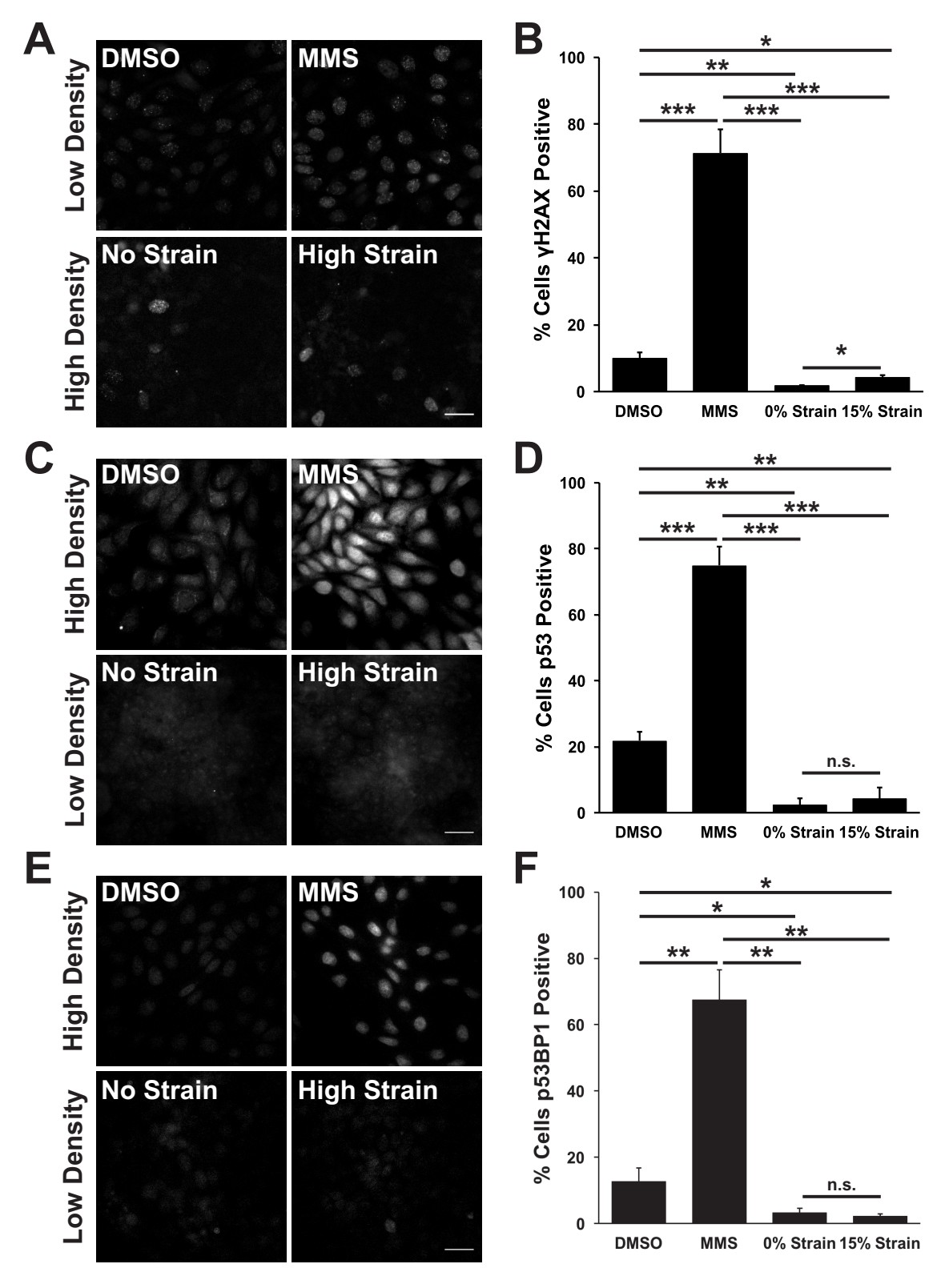

**Figure 2.** Mechanical strain does not induce significant DNA damage or activation of DNA repair. Distribution of phospho-γH2A.X (A), p53 (C), and p53BP1 (E) in MDCK monolayers 24 hr after No Strain or High Strain (15%) using the ISA, or in actively cycling cells treated with DMSO or MMS. Quantification of percent cells phospho-γH2A.X- (B), p53- (D), or p53BP1-positive (F) in each condition. Quantifications were from 3 independent

*Figure 2 continued on next page*

*Figure 2 continued*

experiments and included analysis of 397–3135 cells per experiment. Quantifications were mean +/- SEM; unpaired t-test p values<0.05 (*), <0.01 (**), and <0.001 (***).

The following source data is available for figure 2:

**Source data 1.** Data used to construct graphs in *Figure 2*.

entry, and not due to the abnormal accumulation of DNA damage and activation of the DNA damage checkpoint.

## Strain-induced β-catenin activation is dependent upon Src-mediated phosphorylation and is hyper-activated following inhibition of Casein Kinase I

Since the accumulation of cells in S/G2 following mechanical strain is unlikely to be explained by a DNA damage mitotic checkpoint, we explored the hypothesis that a mitotic driver might be insufficiently activated. β-Catenin, a transcription factor downstream of the Wnt signaling pathway (*Nelson and Nusse, 2004*; *Aberle et al., 1997*; *Morkel et al., 2003*), has a well-characterized role in cellular proliferation and cell cycle progression, and we showed previously that β-catenin transcriptional activity is induced 6–8 hr following mechanical strain and remains at a constant level thereafter (*Benham-Pyle et al., 2015*; *Tan et al., 2016*).

In normal epithelial cells, β-catenin protein levels are tightly regulated through a combination of sequestration in the cadherin complex at cell-cell contacts and active degradation of cytosolic β-catenin by the proteasome (*Aberle et al., 1997*; *Ikeda et al., 1998*; *Su et al., 2008*). Both processes depend on the phosphorylation state of β-catenin, and we tested whether mechanical strain induced β-catenin transcriptional activity through changes in β-catenin phosphorylation. Although the activities of several tyrosine kinases have been reported to affect β-catenin binding to the cadherin adhesion complex (*Lilien and Balsamo, 2005*), we focused on Src phosphorylation of Y654 β-catenin which has been correlated directly with increased Wnt/β-catenin activity (*van Veelen et al., 2011*; *Kajiguchi et al., 2012*) and activation by tissue compression in vivo (*Whitehead et al., 2008*; *Fernández-Sánchez et al., 2015*; *Brunet et al., 2013*); to block β-catenin degradation in the cytoplasm, we focused on β-catenin phosphorylation by CKI/GSK3β by inhibiting the priming phosphorylation by CKI (*Aberle et al., 1997*; *Amit et al., 2002*; *Liu et al., 2002*). Specifically, monolayers were treated with either the Src inhibitor SU6656 (*Blake et al., 2000*) or the CKI inhibitor D4476 (*Rena et al., 2004*). Then, monolayers were fixed and processed for image analysis 8 hr after application of mechanical strain, when β-catenin transcriptional activity increased (*Benham-Pyle et al., 2015*). An MDCK cell line stably expressing the TOPdGFP reporter (*Maher et al., 2009*) was used to measure β-catenin transcriptional activity.

Mechanical strain of monolayers in the absence of SU6656 induced nuclear accumulation of β-catenin and increased β-catenin transcriptional activity as measured by the appearance of nuclear TOPdGFP (*Figure 3A,B*; *Figure 3—figure supplement 1*, see also [*Benham-Pyle et al., 2015*]). Significantly, SU6656 blocked both the nuclear accumulation and transcriptional activity of β-catenin following mechanical strain (*Figure 3A,B*; *Figure 3—figure supplement 1*). Since β-catenin transcriptional activity is required for G1 to S transition following mechanical strain (*Benham-Pyle et al., 2015*), we also measured DNA replication by EdU incorporation in SU6656-treated monolayers after 24 hr (*Figure 3C,D*). In the absence of SU6656, mechanical strain resulted in a significant increase in EdU-positive cells, as reported previously (*Benham-Pyle et al., 2015*). However, treatment with SU6656 blocked EdU incorporation, indicating that Src activity was required for cell cycle re-entry and progression into S following mechanical strain.

Tyrosine 654 on β-catenin is a known Src phosphorylation site, and levels of pY654 β-catenin were examined by immunofluorescence with a pY654 β-catenin-specific antibody (*Rhee et al., 2007*); the small surface area (0.81 cm$^2$) of the ISA does not provide sufficient cell numbers for biochemical characterization (see Materials and methods). Eight hours after mechanical strain, levels of pY654 β-catenin in the cytoplasm and nucleus had increased significantly in mechanically strained cells (DMSO, control); note that levels of pY654 β-catenin fluorescence at cell-cell contacts appeared to

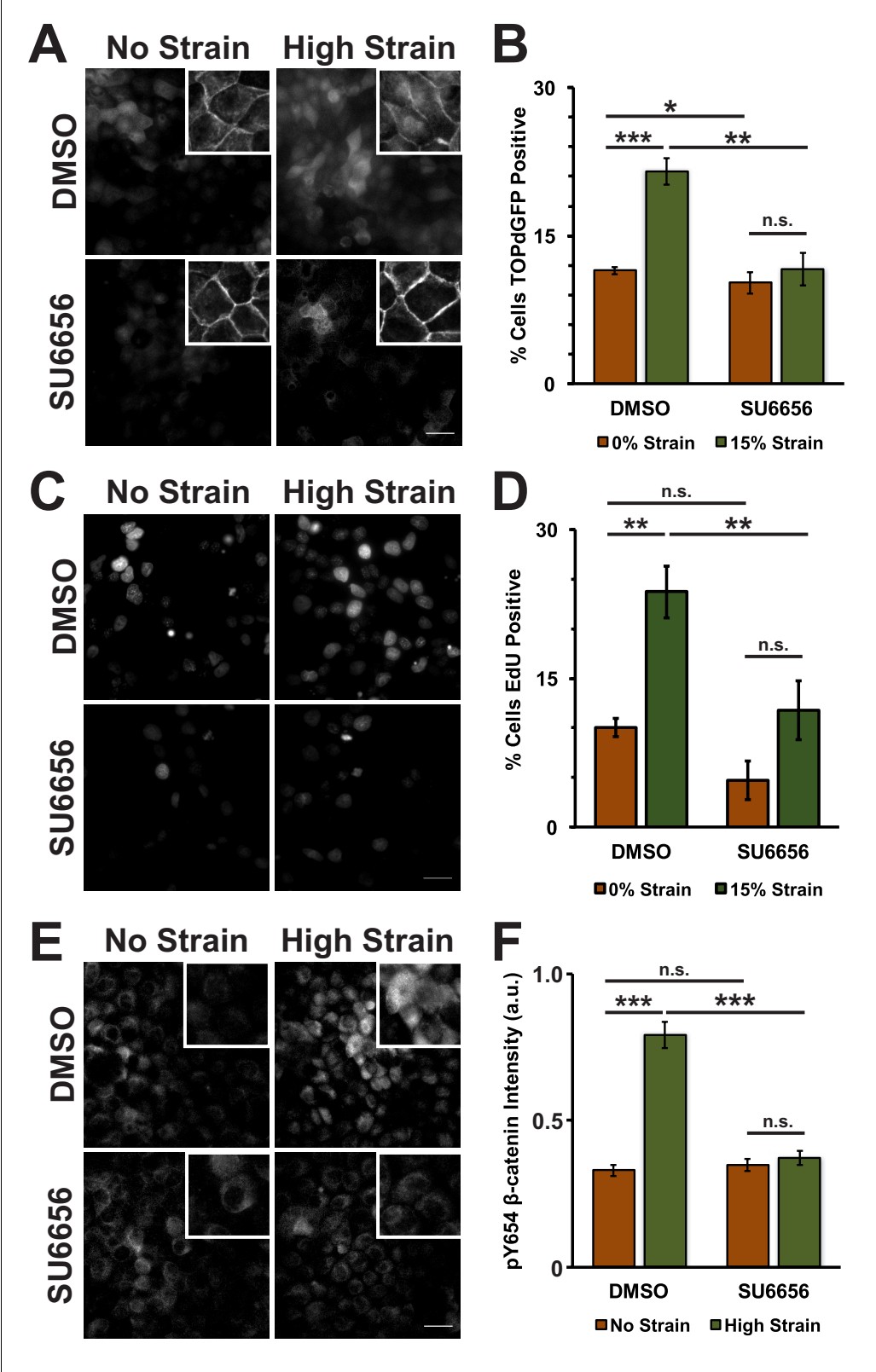

**Figure 3.** Mechanical strain induces a Src-dependent increase in Y654 phosphorylated β-catenin and β-catenin transcriptional activity. Distribution of TOPdGFP at 8 hr (**A**), β-catenin at 8 hr (**A**, insets), EdU at 24 hr (**C**), and pY654 β-catenin at 8 hr (**E**) in MDCK monolayers after No Strain or High Strain (15%) applied by the ISA, treated with either DMSO or the Src inhibitor SU6656 (10 μM). Scale bars: 25 μm. Quantification of percent cells TOPdGFP- (**B**) or EdU- (**D**) positive and quantification of average pY654 β-catenin intensity per pixel (**F**); note that the small surface area (0.81 cm²) of the ISA does

*Figure 3 continued on next page*

*Figure 3 continued*

not provide sufficient cell numbers for biochemical characterization. Quantifications were from at least 3 independent experiments and for the TOPdGFP and EdU quantifications included analysis of 677–1168 cells per experiment. Quantifications were mean +/- SEM; unpaired t-test (**B,D**) or Kolmogorov-Smirnoff (**F**) test p values<0.05 (*), <0.01 (**), and <0.001 (***).

The following source data and figure supplements are available for figure 3:

**Source data 1.** Data used to construct graphs in *Figure 3* and *Figure 3—figure supplement 1*.

**Figure supplement 1.** Distribution of β-catenin in monolayers treated with either DMSO or the Src Inhibitor SU6656 8 hr after the application of No Strain or High Strain (15%) using the ISA.

**Figure supplement 2.** The Src Inhibitor PP2 blocks strain-induced increases in Y654 phosphorylated β-catenin and β-catenin transcriptional activity.

**Figure supplement 3.** Distribution of β-catenin in monolayers treated with either DMSO or the Src Inhibitor PP2 8 hr after the application of No Strain or High Strain (15%) using the ISA.

**Figure supplement 4.** EGFR inhibition reduces an increase in pY654 β-catenin following mechanical strain, but does not affect β-catenin transcriptional activity or cell cycle re-entry.

**Figure supplement 5.** Distribution of β-catenin in monolayers treated with either DMSO or the EGFR Inhibitor PD153035 8 hr after the application of No Strain or High Strain (15%) using the ISA.

be low, as expected since pY654 β-catenin has a weaker affinity for binding E-cadherin (*Coluccia et al., 2007*; *Huber and Weis, 2001*; *Zeng et al., 2006*). However, treatment with SU6656 resulted in a complete block of this strain-induced increase in Y654 β-catenin, which remained at levels similar to the unstrained control (*Figure 3E,F*). Similar results to those with SU6656 treatment were obtained upon treatment with the tyrosine kinase inhibitor PP2 (*Hanke et al., 1996*), which blocked increases in pY654 β-catenin, nuclear localization and transcriptional activity of β-catenin, and EdU incorporation following mechanical strain (*Figure 3—figure supplement 2* and *3*).

Increased pY654 β-catenin levels have been correlated with activation of Epidermal Growth Factor Receptor (EGFR) signaling (*Lilien and Balsamo, 2005*; *Shibata et al., 1996*; *Hazan and Norton, 1998*). To test whether EGFR activation was involved in strain-induced accumulation of pY654 β-catenin and β-catenin signaling, monolayers were treated with the EGFR inhibitor PD153035 (*Fry et al., 1994*; *Bos et al., 1997*). EGFR inhibition did not significantly affect strain-induced increases in either β-catenin transcriptional activity or cell cycle re-entry (*Figure 3—figure supplement 4A–D* and *5*). Inhibition of EGFR by PD153035 reduced the increase in pY654 β-catenin following mechanical strain by ~40% (*Figure 3—figure supplement 4E,F*), unlike the complete inhibition of strain-induced pY654 β-catenin accumulation by the Src inhibitor SU6656 (*Figure 3E,F*). These results indicate that EGFR activation may contribute to the increase in pY654 β-catenin following mechanical strain (*Muhamed et al., 2016*) but it is not required for strain-induced pY654 β-catenin-mediated activation of cell cycle progression.

To test whether inhibition of CKI phosphorylation affected β-catenin transcriptional activity in response to mechanical strain, monolayers were treated with the CKI inhibitor D4476 (*Rena et al., 2004*). D4476 resulted in an increase in cytoplasmic and nuclear β-catenin in the absence of mechanical strain, but a significantly greater increase in cytoplasmic and nuclear β-catenin after the application of mechanical strain (*Figure 4A*; *Figure 4—figure supplement 1*). Similarly, D4476 caused an increase in levels of β-catenin transcriptional activity indicated by the TOPdGFP reporter system in unstrained monolayers, but a significantly greater increase in mechanically strained monolayers (*Figure 4B*). D4476 treatment also increased EdU incorporation in combination with mechanical strain, indicating that CKI inhibition increased the probability of cells progressing from G1 into S. D4476 treatment did not increase EdU incorporation in un-strained monolayers (*Figure 4C,D*), indicating that CKI inhibition alone did not increase cell cycle re-entry from quiescence, which requires mechanical strain and Yap1 transcriptional activity (*Benham-Pyle et al., 2015*). D4476 treated monolayers were also stained for pY654 β-catenin to test whether CKI inhibition effected the accumulation

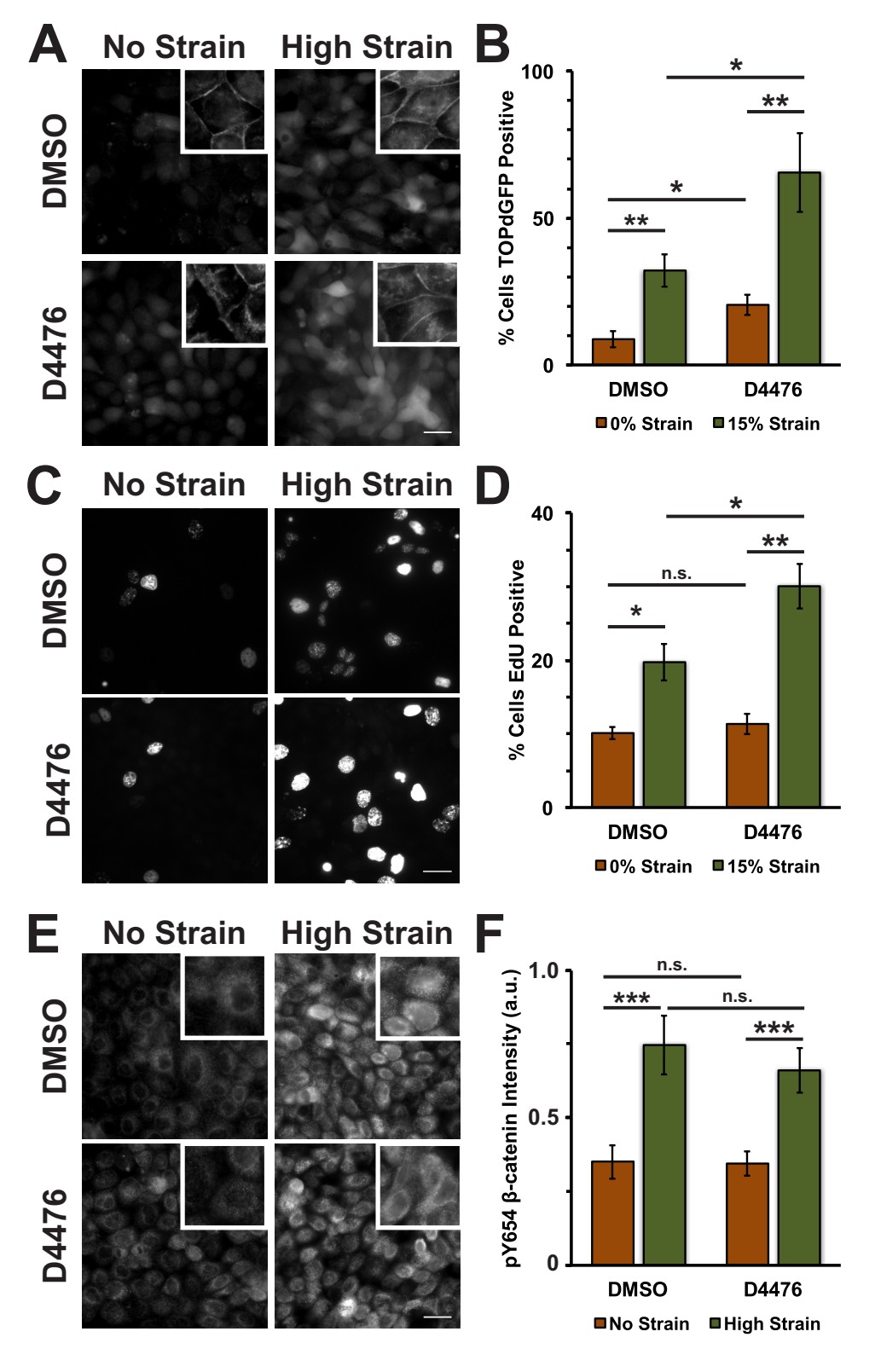

**Figure 4.** Inhibition of Casein Kinase I (CKI) increases β-catenin transcriptional activity in MDCK quiescent monolayers, independent of mechanical strain. Distribution of TOPdGFP at 8 hr (A), β-catenin at 8 hr (A, insets), EdU at 24 hr (C), and pY654 β-catenin at 8 hr (E) in MDCK Monolayers after No Strain or High Strain (15%) applied by the ISA and treated with either DMSO or the CKI inhibitor D4476. Quantification of percent cells TOPdGFP- (B) or EdU- (D) positive and average pY654 β-catenin intensity per pixel (F). Quantifications were from at least 3 independent experiments and, for the

*Figure 4 continued on next page*

*Figure 4 continued*

TOPdGFP and EdU quantifications, included analysis of 832–1364 cells per experiment. Quantifications were mean +/- SEM; unpaired t-test (**B**, **D**) or Kolmogorov-Smirnoff (**F**) test p values<0.05 (*), <0.01 (**), and <0.001 (***).

The following source data and figure supplement are available for figure 4:

**Source data 1.** Data used to construct graphs in *Figure 4* and *Figure 4—figure supplement 1*.

**Figure supplement 1.** Distribution of β-catenin in monolayers treated with either DMSO or the CKI Inhibitor D4476 8 hr after the application of No Strain or High Strain (15%) using the ISA.

of pY654 β-catenin following mechanical strain. Significantly, CKI inhibition did not appear to change the level of pY654-β-catenin before or after mechanical strain compared to the control (DMSO) (*Figure 4E,F*). These results indicate that mechanical strain-induced changes in the phosphorylation state of β-catenin by Src and modulation of β-catenin degradation by CKI regulate strain-induced increases in cytoplasmic β-catenin and β-catenin trancriptional activity.

## Inhibition of β-catenin degradation promotes cell cycle progression through mitosis following mechanical strain

Mechanical strain induced cell cycle re-entry and accumulation of cells in S/G2 but was insufficient to drive cells into mitosis, suggesting an additional level of regulation for cell transition from S/G2 to mitosis in the presence of mechanical strain. Since β-catenin transcriptional activity is required for progression through S phase, we considered whether further increasing the level of β-catenin activity, through inhibition of β-catenin degradation, might be sufficient to drive cells into, and through mitosis in the presence of mechanical strain.

D4476-treated Fucci-MDCK monolayers were imaged on the biaxial live imaging cell stretcher for 24 hr following application of mechanical strain. In the absence of mechanical strain, the number of Geminin-positive cells in S/G2 in both control (DMSO) and D4476 treated monolayers was very low and did not change over 24 hr (*Figure 5A,B*, *Video 3*). In contrast to control monolayers, in which the number of Geminin-positive cells increased gradually after 6–8 hr (*Figure 5A,C*, *Video 1*), D4476 treatment resulted in a faster accumulation of Geminin-positive cells, resulting in a significant increase in the number of cells in S/G2 after 24 hr compared to the DMSO control (*Figure 5A,C*, *Video 4*). Single-cell tracking of Geminin-positive cells showed an increase in cells transitioning from G1 to S (red to green fluorescence, *Figure 5D*) in mechanically strained monolayers treated with D4476 compared to monolayers treated with either D4476 or mechanical strain alone.

To assess cell entry into mitosis, Fucci MDCK cells were tracked and the number of mitotic events per hour was measured as a function of the degradation of Geminin and cells switching from green to red fluorescence (*Figure 6A*, *Videos 1–6*). In unstrained monolayers, D4467 treatment increased the level of mitotic events beginning 6 hr after treatment and this level then gradually decreased to approximately the control level over 24 hr (*Figure 6B*, *Video 3*). Mechanically strained monolayers treated with D4476 had increased numbers of mitotic events after 5 hr, which continued to increase and remained significantly elevated over 24 hr (*Figure 6C*, *Video 4*). Single-cell tracking of Geminin-positive cells confirmed that the number of cells that transitioned from G1 into S and then divided (red to green to red fluorescence, *Figure 6A*, white arrows) increased in mechanically strained monolayers treated with D4476, but not in the presence of D4476 or mechanical strain alone (*Figure 6D*).

CKI contributes in multiple ways to cell cycle progression and directly to mitotic entry (*Meisner and Czech, 1991*; *Brockman et al., 1992*; *Hanna et al., 1995*; *Ho et al., 1997*). Therefore, it was important to test whether the effect of CKI inhibition on mechanical strain-induced entry of cells into mitosis required β-catenin transcriptional activity. Previously, we showed that iCRT3, a small molecule inhibitor of β-catenin/TCF interactions (*Gonsalves et al., 2011*), completely blocked β-catenin transcriptional activity and transition of cells from G1 into S following mechanical strain (*Benham-Pyle et al., 2015*). Significantly, addition of both D4467 and iCRT3 to mechanically strained monolayers blocked the increase in Geminin-positive cells following mechanical strain and the number of cells entering mitosis observed with D4476 alone (*Figures 5* and *6*, *Videos 5* and *6*). These

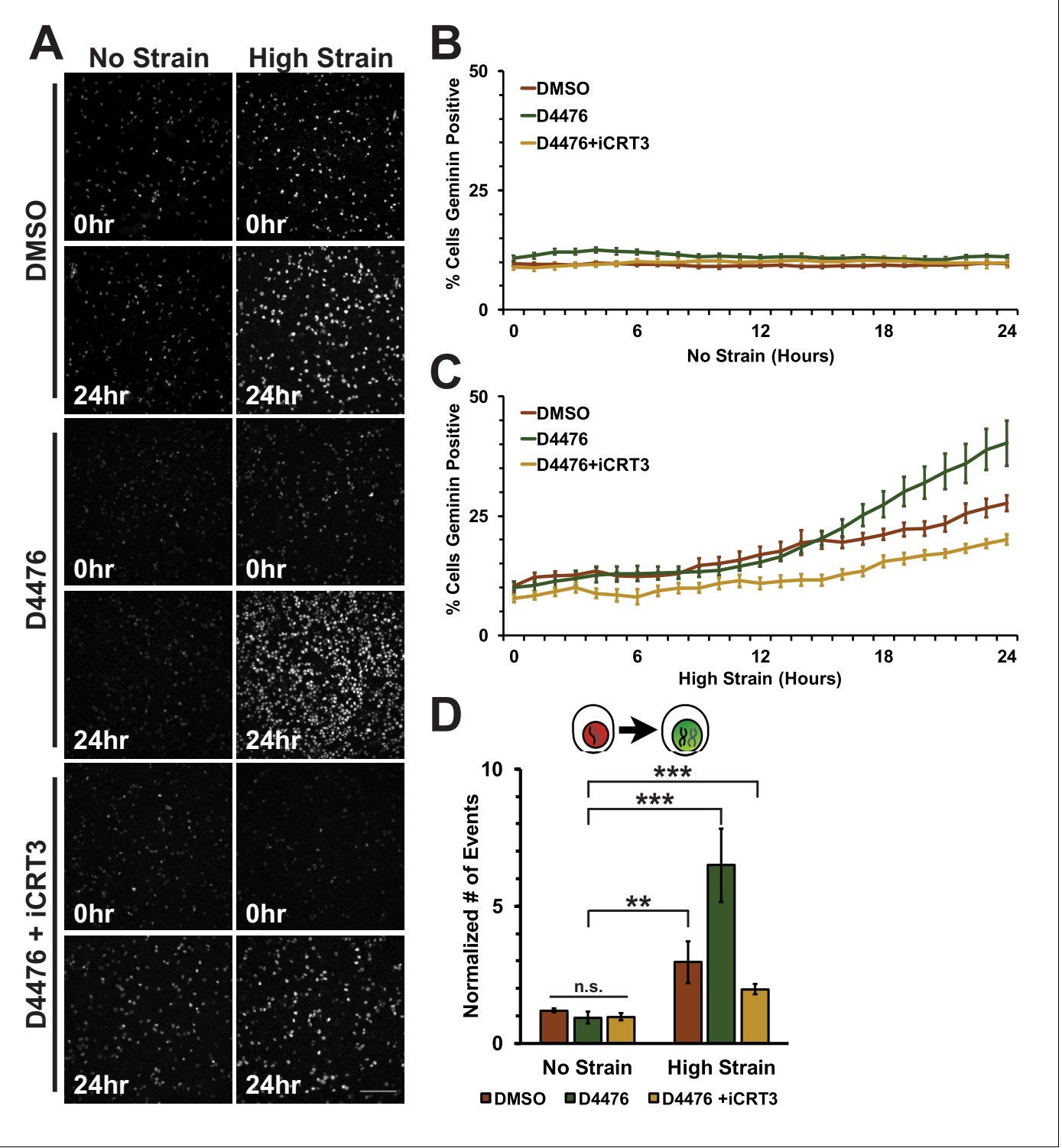

**Figure 5.** Inhibition of β-catenin degradation promotes progression from G1 into S/G2 following mechanical strain. (**A**) Distribution of mAG-Geminin in Fucci MDCK monolayers treated with DMSO, D4476, or D4476 and iCRT3 0 hr or 24 hr after the application of No Strain or High Strain (~8.5%) using the biaxial live cell stretcher. Scale bar: 150 μm. Quantification of percent cells Geminin positive in Fucci-MDCK monolayers after No Strain (**B**) or High Strain (15%) (**C**); percent cells Geminin positive in D4476 treated monolayers are statistically significant (p<0.05) relative to DMSO monolayers following mechanical strain at 16–24 hr. (**D**) Single cell tracking quantification of number of cell objects accumulated in S/G2 (red to green fluorescence, left) during 24 hr. All quantifications were from 2–4 independent experiments and included analysis of at least 15,000 cells. Quantifications were mean +/- SEM; unpaired t-test (**B, C**) or Kolmogorov-Smirnoff (**D**) test p values<0.01 (**), and <0.001 (***).

*Figure 5 continued on next page*

*Figure 5 continued*

The following source data is available for figure 5:

**Source data 1.** Data used to construct graphs in *Figure 5*.

results indicate that the effects of CKI inhibition on cell cycle progression and mitotic entry involve increased β-catenin transcriptional activity.

## Mechanical strain induces cell cycle re-entry and mitosis when combined with Wnt3A stimulation

In vivo, Wnt signaling blocks β-catenin degradation, resulting in β-catenin accumulation in the cytoplasm and nucleus and induction of β-catenin transcriptional activity (*Clevers and Nusse, 2012*; *Korinek et al., 1997*; *Clevers, 2006*; *Chen et al., 2012*; *Chen et al., 2001*; *Reya and Clevers, 2005*). Therefore, we tested if β-catenin stabilization and transcriptional activation by Wnt, similar to CKI inhibition, could act synergistically with mechanical strain to drive cell cycle progression and entry into mitosis.

Conditioned media was collected from control L cells or L cells expressing Wnt3A, as previously described (*Lyons et al., 2004*). Both unstrained and strained monolayers treated with Wnt3A conditioned media had high levels of nuclear TOPdGFP compared to similarly strained monolayers treated with control conditioned medium (*Figure 7—figure supplement 1*). Mechanical strain in the absence of Wnt3A stimulation induced an increase in Geminin-positive S/G2 cells after an 8-hour delay compared to unstrained monolayers(*Figure 7A*; *Figure 7C*, *Video 7* and *Video 8*), similar to results observed with DMSO treated monolayers. Despite the increase in nuclear TOPdGFP in monolayers treated with Wnt3A conditions media, the number of Geminin-positive cells in unstrained monolayers in the presence of control or Wnt3A condition medium was similarly low (*Figure 7A,B*, *Videos 7* and *Video 9*), indicating that in the absence of mechanical strain Wnt3A and increased β-catenin transcription activity were insufficient to drive cells into S; note that mechanical strain-induced Yap1 transcriptional activity is required first for quiescent cell re-entry into G1 (and hence progression to S), even if β-catenin transcription activity is high (*Benham-Pyle et al., 2015*).

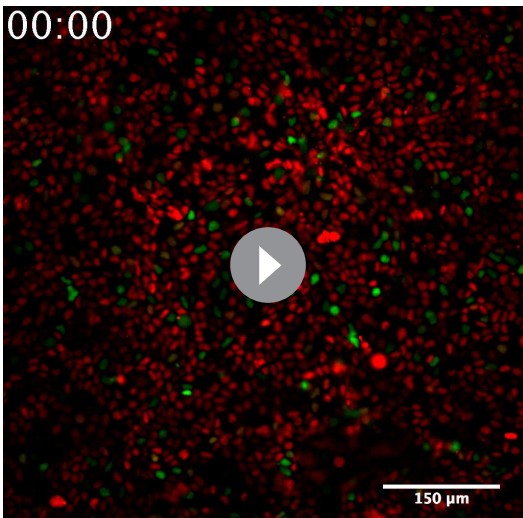

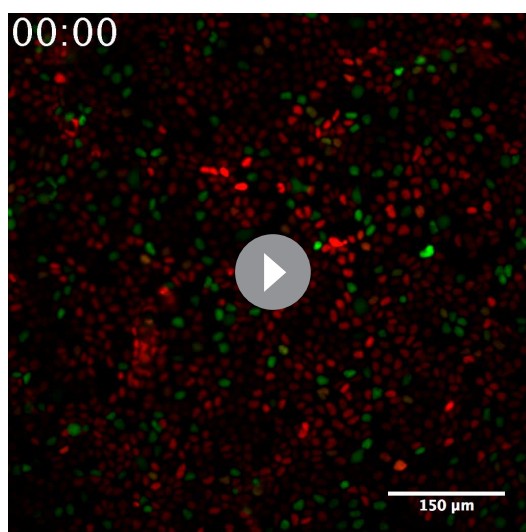

**Video 3.** Representative movie of a Fucci-MDCK monolayer treated with D4476 after no strain. Same data as in *Figures 5* and *6*.

**Video 4.** Representative movie of an FUCCI-MDCK monolayer treated with D4476 after mechanical strain. Same data as in *Figures 5* and *6*.

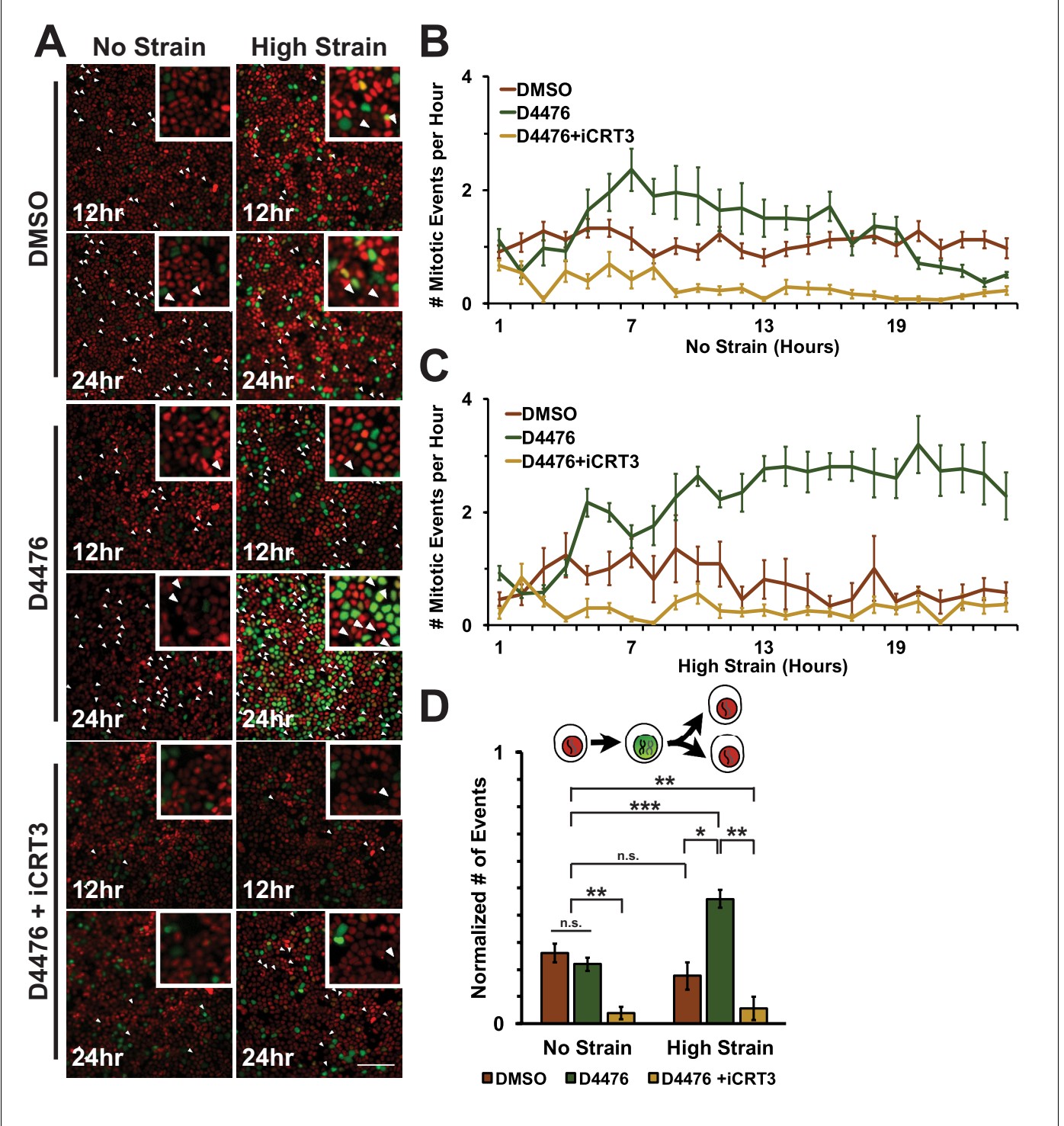

**Figure 6.** Inhibition of β-catenin degradation promotes mitotic entry following mechanical strain. (A) Distribution of mitotic events (white arrow heads) in Fucci MDCK monolayers 12 hr or 24 hr after the application of No Strain or High Strain (~8.5%) using the biaxial live cell stretcher and treatment with DMSO, D4476, or D4476 and iCRT3. Scale bar: 100 μm. (B) Number of mitotic events per hour in treated Fucci-MDCK monolayers following No Strain; mitotic events in D4476 treated monolayers were statistically significant (p<0.05) compared to DMSO treated monolayers from 6–16 hr, and at 22 and 24 hr (C) Number of mitotic events per hour in treated Fucci-MDCK monolayers following High Strain; mitotic events in D4476 treated monolayers were statistically significant (p<0.05) when compared to DMSO treated monolayers from 5–24 hr. (D) Single cell tracking quantification of number of cell objects that passed through S/G2 and divided (red to green to red fluorescence, right) during 24 hr. All quantifications were from 2–4 independent

*Figure 6 continued on next page*

*Figure 6 continued*

experiments and included analysis of at least 15,000 cells. Quantifications were mean +/- SEM; Kolmogorov-Smirnoff test p values<0.05 (*), <0.01 (**), and <0.001 (***).

The following source data is available for figure 6:

**Source data 1.** Data used to construct graphs in *Figure 6*.

Significantly, monolayers under mechanical strain and treated with Wnt3A-conditioned media accumulated more Geminin-positive S/G2 cells than mechanically strained monolayers in the presence of control conditioned media (*Figure 7A, C*, *Video 10*). Moreover, single-cell tracking of Geminin-positive cells showed an increase in cells transitioning from G1 to S (red to green fluorescence, *Figure 7D*, *Videos 7–10*) in mechanically strain monolayers treated with Wnt3A-conditioned media, compared to monolayers treated with either Wnt3A-conditioned media or mechanical strain alone.

In unstrained monolayers, Wnt3A stimulation increased the number of mitotic events over 5 hr, which then gradually decreased over 24 hr (*Figure 8B*, *Video 7*). Mechanically strained monolayers treated with Wnt3A also had an initial peak in the number of mitotic events at ~5 hr, but, in contrast to the control, the number of mitotic events increased significantly after ~8 hr and continued to increase over 24 hr (*Figure 8C*, *Video 10*). Single-cell tracking of Geminin-positive cells confirmed that the number of cells that transitioned from G1 into S and then divided (red to green to red fluorescence, *Figure 8A*, white arrows) increased in mechanically strained monolayers stimulated with Wnt3A, but not in the presence of either Wnt3A or mechanical strain alone (*Figure 8D*, *Videos 7–10*). Note that these effects of Wnt3A in driving mechanically strained cells into mitosis were very similar to those induced by D4467 (compare *Figures 6* and *8*).

## Discussion

Mechanical cues are critical for the normal development, morphology, and function of multicellular tissues. Numerous pathways and molecular scaffolds have been identified as mechano-responsive, including cadherin cell-cell adhesion complexes (*Liu et al., 2010*; *Cai et al., 2014*), integrin-mediated focal adhesions (*Wang et al., 2001*), the

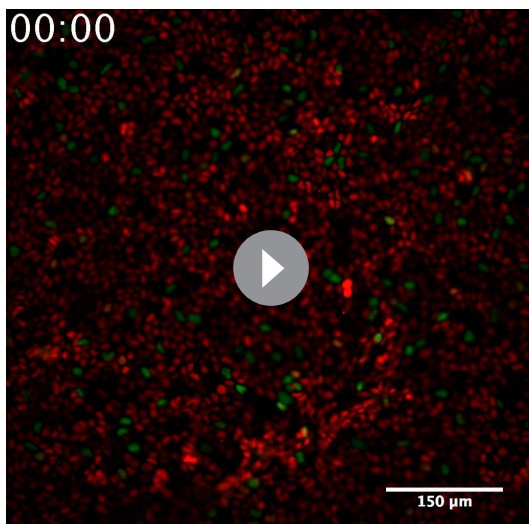

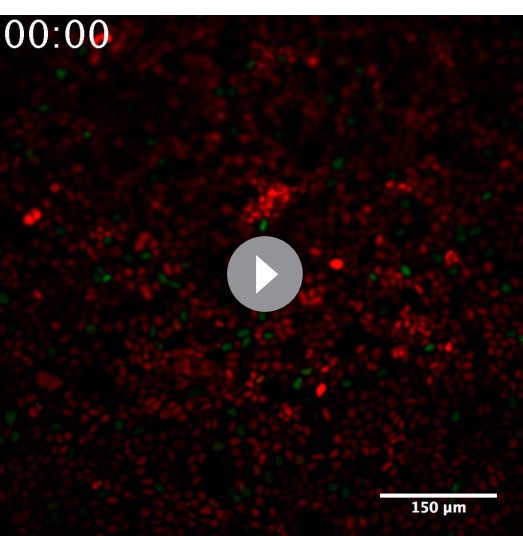

**Video 5.** Representative movie of a Fucci-MDCK monolayer treated with D4476 and iCRT3 after no strain. Same data as in *Figures 5* and *6*.

**Video 6.** Representative movie of a Fucci-MDCK monolayer treated with D4476 and iCRT3 after mechanical strain. Same data as in *Figures 5* and *6*.

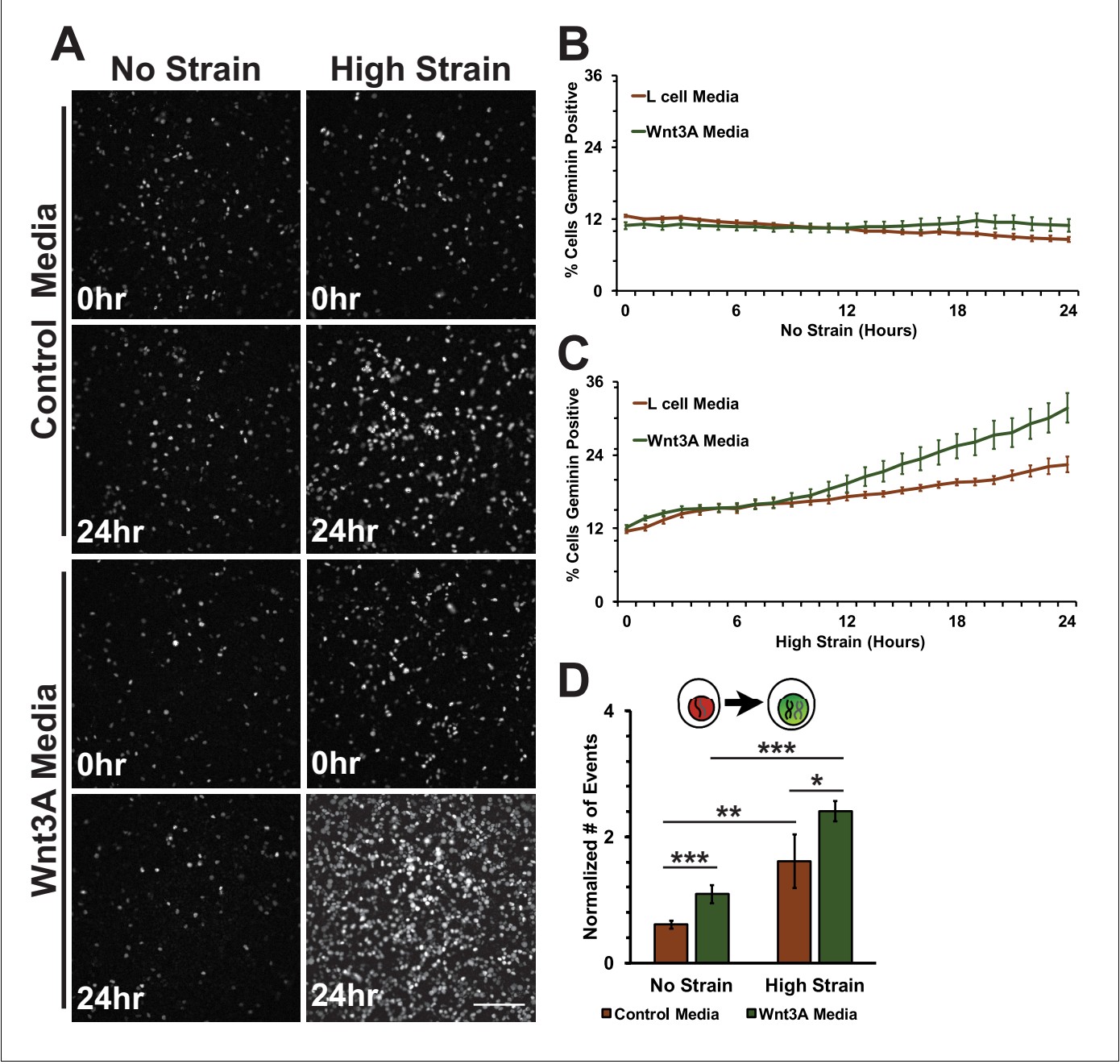

**Figure 7.** Mechanical strain induced increase in the progression from G1 into S/G2 in the presence of Wnt3A. (A) Distribution of mAG-Geminin in Fucci MDCK monolayers treated with control or Wnt3A-conditioned media 0 hr or 24 hr after the application of No Strain or High Strain (~8.5%) using the biaxial live cell stretcher. Scale bar: 150 µm. Quantification of percent cells Geminin positive in Fucci-MDCK monolayers after No Strain (B) or High Strain (15%) (C); percent cells Geminin positive in Wnt3A treated monolayers were statistically significant (p<0.05) relative to control monolayers following mechanical strain at 14–24 hr. (D) Single cell tracking quantification of number of cell objects accumulated in S/G2 (red to green fluorescence, left) during 24 hr. All quantifications were from 2–3 independent experiments and included analysis of at least 15,000 cells. Quantifications were mean +/- SEM; unpaired t-test (B, C) or Kolmogorov-Smirnoff (D) test p values<0.05 (*), <0.01 (**), and <0.001 (***).

The following source data and figure supplement are available for figure 7:

**Source data 1.** Data used to construct graphs in *Figure 7* and *Figure 7—figure supplement 1*.

**Figure supplement 1.** Wnt3A induces increased β-catenin transcriptional activity, independent of mechanical strain.

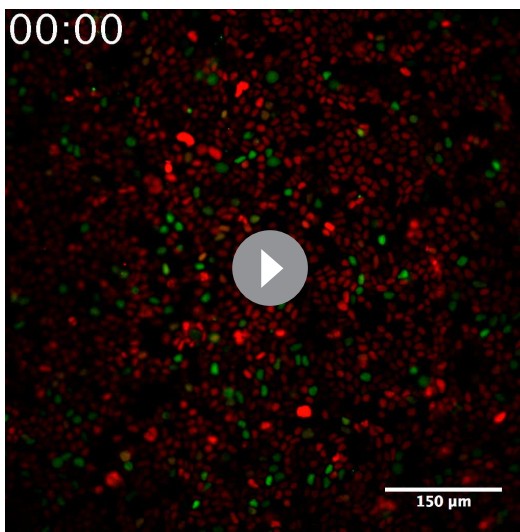

**Video 7.** Representative movie of a Fucci-MDCK monolayer treated with control-conditioned media after no strain. Same data as in *Figures 7* and *8*.

**Video 8.** Representative movie of a Fucci-MDCK monolayer treated with control-conditioned media after mechanical strain. Same data as in *Figures 7* and *8*.

actin cytoskeleton (*Hoffman et al., 2011*; *Engl et al., 2014*), and Yap1 and β-catenin (*Whitehead et al., 2008*; *Benham-Pyle et al., 2015*; *Aragona et al., 2013*; *Kamel et al., 2010*; *Dupont et al., 2011*; *Heo and Lee, 2011*). Whether mechanical cues synergize with signaling pathways known to regulate development and homeostasis remains a relatively unexplored area of investigation. Previously, we reported that mechanical strain sequentially induced Yap1 and β-catenin transcriptional activity to drive cell cycle re-entry (Yap1) and progression from G1 into S (β-catenin) (*Benham-Pyle et al., 2015*). Here, we show that these cells are held in S/G2 and do not enter mitosis, and that further activation of the Wnt signaling pathway and β-catenin transcriptional activity are required to drive cell cycle progression through mitosis.

The accumulation of mechanically strained cells in S/G2 did not appear to be caused by the activation of the DNA damage checkpoint, as the level of DNA damage and DNA repair pathway activation in those cells, measured by levels of phospho-γH2A.X, p53 and p53BP1, was insignificant compared to that induced by a DNA damaging agent and present in actively cycling cells (*Figure 2*). In vivo, G2 delay prior to mitosis has been identified in select developmental niches, including early stages of *Drosophila* development (*Edgar, 1990*), the *Drosophila* zone of non-proliferating cells (ZNC) (*Johnston and Edgar, 1998*), and during *Drosophila* neural fate determination (*Nègre et al., 2003*). Our results indicate that quiescent cells may have evolved regulatory mechanisms to arrest healthy cells in S/G2 under circumstances unfavorable for mitosis; for example, mitosis at high cell density might result in the extrusion of cells and the disruption of monolayer integrity.

β-Catenin is a well-characterized regulator of cell cycle progression (*Nelson and Nusse, 2004*; *Orford et al., 1999*; *Morkel et al., 2003*; *Olmeda et al., 2003*), and is responsive to mechanical cues (*Farge, 2003*; *Benham-Pyle et al., 2015*; *Brunet et al., 2013*). While mechanical strain-induced activation of β-catenin transcriptional activity requires E-cadherin *trans* adhesion interactions (*Benham-Pyle et al., 2015*), it remains unclear whether mechanical strain induces β-catenin transcriptional activity through destabilizing the junctional pool of cadherin-bound β-catenin, or stabilizing cytoplasmic β-catenin, or a combination of both.

Here, we showed that mechanical strain induced a Src-dependent increase in tyrosine phosphorylated (pY654) β-catenin in the cytoplasm and nucleus. pY654 β-catenin has a weaker affinity for E-cadherin (*Huber and Weis, 2001*; *Zeng et al., 2006*), and thus may be released from the cadherin complex into the cytoplasm as indicated by the increased cytoplasmic staining of pY654 β-catenin in mechanically strained monolayers (*Figure 3E*). Importantly, SU6656 inhibition of Src activity completely blocked the

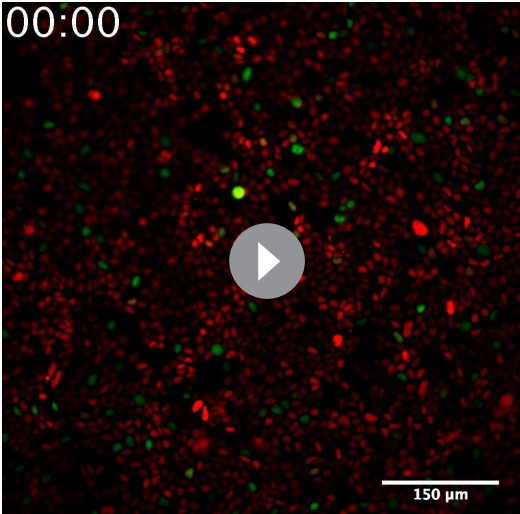

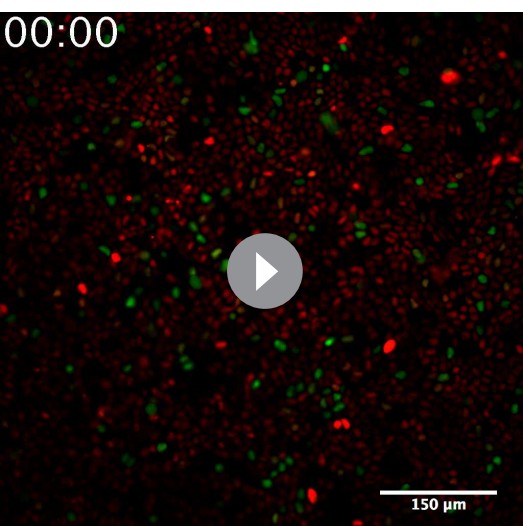

**Video 9.** Representative movie of a Fucci-MDCK monolayer treated with Wnt3A-conditioned media after no strain. Same data as in *Figures 7* and *8*.

**Video 10.** Representative movie of a Fucci-MDCK monolayer treated with Wnt3A-conditioned media after mechanical strain. Same data as in *Figures 7* and *8*.

strain-induced increases in Y654 β-catenin phosphorylation, β-catenin accumulation in the cytoplasm and nucleus, β-catenin transcriptional activity, and DNA replication (*Figure 3*). Nevertheless, we also tested the possibility that other tyrosine kinases were involved in cellular responses to mechanical strain (*Lilien and Balsamo, 2005*; *Tamada et al., 2012*; *Bays et al., 2014*). Indeed, inhibition of EGFR partially reduced the strain-induced increase in level of pY654 β-catenin, but did not significantly affect the strain-induced increase in β-catenin transcriptional activity or cell cycle re-entry (*Figure 3—figure supplement 4* and *5*). Since Src inhibition blocked strain-induced pY654 β-catenin accumulation, β-catenin transcriptional activity and cell cycle progression, we conclude that Src activation plays the predominant role in activating β-catenin signaling in response to mechanical strain. How Src is activated by strain is currently unknown. However, our earlier finding that E-cadherin *trans* adhesion interactions are required for mechanical strain-induced β-catenin transcriptional activity and cell cycle progression (*Benham-Pyle et al., 2015*) indicates that Src activation may be downstream of E-cadherin extracellular engagement.

In contrast to the effects of inhibiting Src activity, inhibition of CKI activity increased β-catenin transcriptional activity (*Figure 4*) independently of mechanical strain. Normally, CKI phosphorylation of β-catenin is the first step in targeting cytosolic β-catenin for degradation (*Amit et al., 2002*; *Maher et al., 2009*). Since CKI inhibition did not change the level of pY654 β-catenin following mechanical strain (*Figure 4E,F*), the effects of CKI inhibition and mechanical strain likely act through different biochemical pathways.

Live imaging of Fucci-MDCK monolayers revealed that increased β-catenin transcriptional activity upon CKI inhibition produced a distinct cell cycle response to mechanical strain compared to untreated monolayers (*Figures 5* and *6*). Following mechanical strain, CKI inhibition increased both the number of cells that accumulated in S/G2 and the number of mitotic events. However, only cells exposed to both CKI inhibition and mechanical strain re-entered the cell cycle and divided within 24 hr. Importantly, this increase in mitotic events was dependent on β-catenin transcriptional activity. Components of the β-catenin degradation machinery are commonly mutated in tumors (*Morin et al., 1997*; *Bottomly et al., 2010*; *Mishra et al., 2015*), and inactivation of β-catenin degradation is thought to be an important event in tumor development. However, it has remained unclear what additional factors may be important for metastasis and malignant growth. Our results indicate that mechanical strain may trigger accelerated growth within a mutation-induced, pre-malignant background of increased β-catenin transcriptional activity.

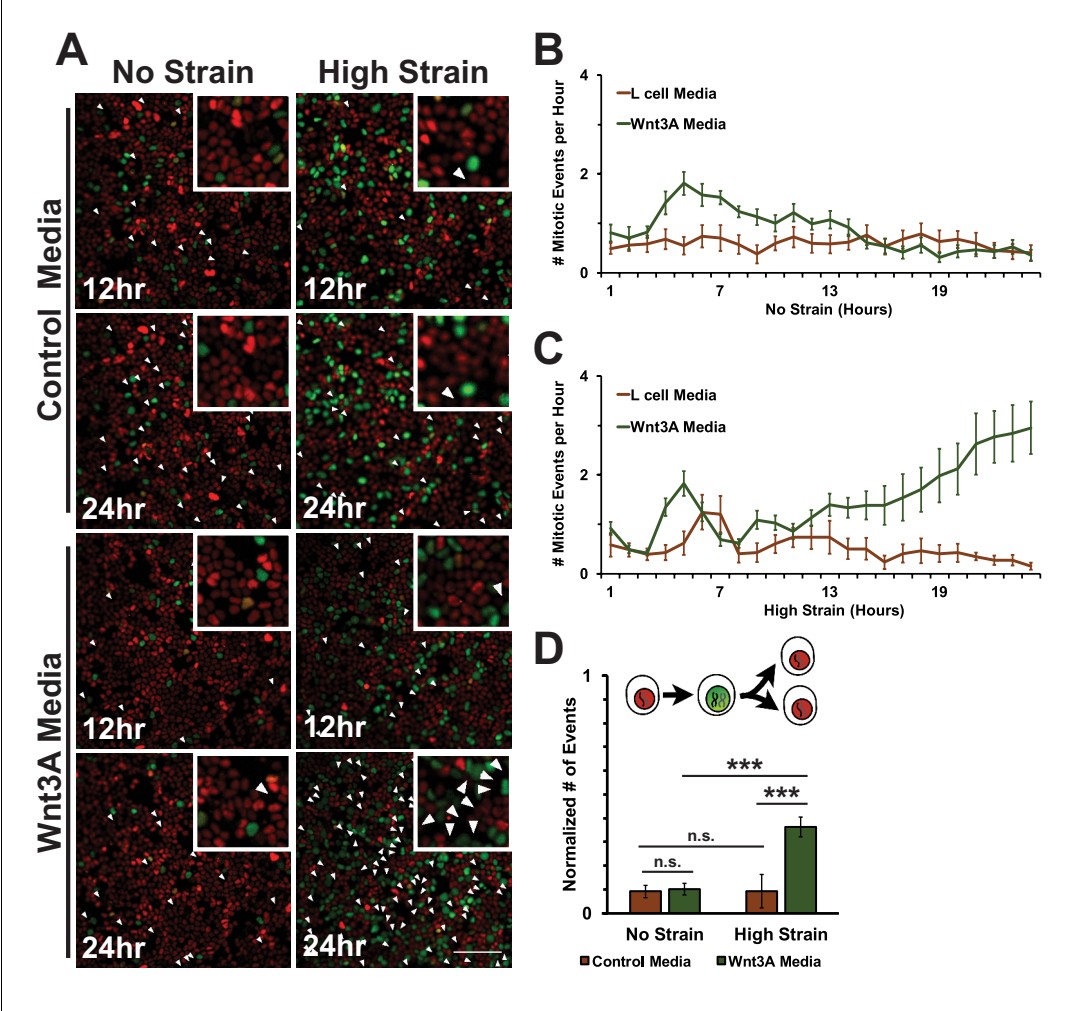

**Figure 8.** Mechanical strain induced increased mitotic entry in the presence of Wnt3A. (A) Distribution of mitotic events (white arrow heads) in Fucci MDCK monolayers 12 hr or 24 hr after the application of No Strain or High Strain (~8.5%) using the biaxial live cell stretcher in the presence of control or Wnt3A-conditioned media. Scale bar: 100 µm. (B) Number of mitotic events per hour in treated Fucci-MDCK monolayers following No Strain; mitotic events in Wnt3A treated monolayers were statistically significant (p<0.05) compared to control monolayers from 4–14 hr (C) Number of mitotic events per hour in treated Fucci-MDCK monolayers following High Strain; mitotic events in Wnt3A treated monolayers were statistically significant (p<0.05) compared to control monolayers from 4–24 hr. (D) Single cell tracking quantification of number of cell objects that passed through S/G2 and divided (red to green to red fluorescence, right) during 24 hr. All quantifications were from 2–3 independent experiments and included analysis of at least 15,000 cells. Quantifications were mean +/- SEM; Kolmogorov-Smirnoff test p values<0.001 (***).
The following source data is available for figure 8:

**Source data 1.** Data used to construct graphs in *Figure 8*.

CKI inhibition is a simple method to stabilize cytoplasmic β-catenin by short-circuiting the β-catenin degradation machinery, but it has the potential for off-target effects given the broad functions of CKI in cells. Wnt ligands are secreted signaling proteins that induce β-catenin transcriptional activity by inhibiting β-catenin phosphorylation by CKI/GSK3β (*Li et al., 2012*; *Vinyoles et al., 2014*). In the absence of mechanical strain, Wnt3A stimulation of quiescent MDCK monolayers was not sufficient to drive cell cycle re-entry or an accumulation of cells in S/G2. Note also that levels of pY654 β-catenin were also unchanged in the presence of Wnt3A (*Figure 7—figure supplement 1*), similar to CKI inhibition (*Figure 4E,F*), indicating a Wnt-independent pathway for Src activation. Upon mechanical strain, Wnt3A-stimulated monolayers had significantly higher numbers of cells both in S/G2 and entering mitosis compared to monolayers treated with Wnt3A or mechanical strain alone (*Figures 7*

and *8*). The linear increase in mitotic events after an 8–10 hr delay following mechanical strain in the presence of Wnt3A is consistent with the hypothesis that monolayers exposed to both mechanical strain and Wnt3A exited quiescence (G0), proceeded through DNA replication and then divided without a delay in S/G2. This is in sharp contrast to mechanically strained monolayers without Wnt3A stimulation that accumulated in S/G2.

Dose-dependent effects and signaling thresholds have been previously described in early development (*Green and Smith, 1990*; *Green et al., 1992*; *Zhang et al., 2013*), and several signaling pathways, including Wnt/β-catenin signaling, have been suggested to function as continuously variable signaling pathways (*Hazzalin and Mahadevan, 2002*; *Bardwell, 2008*; *Goentoro and Kirschner, 2009*). Our results showed that mechanical strain or Wnt3A addition activated β-catenin signaling to a level sufficient to drive some cells through G1 into S (*Figure 1*, *Figure 7D*). However, the combination of mechanical strain and Wnt3A addition induced higher β-catenin signaling levels (*Figure 7—figure supplement 1*), which drove more cells through G1 into S (*Figure 7*) and through G2 into mitosis (*Figure 8*). This synergistic effect of Wnt3A and mechanical strain on both β-catenin transcriptional activity and cell cycle progression confirms that Wnt/β-catenin activity is not a stable binary switch, but rather involves a graded activation in which different levels of β-catenin transcriptional activity produce different cell cycle responses (lower for G1 to S/G2, and higher for S/G2 into mitosis).

It has been a long-standing hypothesis that growth-induced local mechanical cues could influence morphogenesis by modulating growth rates, the direction of growth, or differentiation (*Henderson and Carter, 2002*). Indeed, a model integrating both force-sensing and signaling pathways accounted for experimentally observed patterns of growth, cell shape, and cell size In the *Drosophila* wing imaginal disc (*Aegerter-Wilmsen et al., 2012*). This model was supported by the observation that a global pattern of stress in the wing imaginal disc coincided with polarized actomyosin contractility in the cell cortex and the alignment of the division plane with the main axis of cell stretch, thereby contributing to tissue elongation. However, very few studies have demonstrated how mechanical forces could synergize with signaling pathways to effect growth rates.

How might Wnt signaling and mechanical forces synergize in vivo? Canonical Wnt signaling can act as a locally restricted morphogen to regulate limb development, asymmetric stem cell divisions, and sensitivity to other effector pathways during development and adult tissue homeostasis. (*Ng et al., 1996*; *Habib et al., 2013*; *Lindsley et al., 2006*; *Alexandre et al., 2014*). In mammary and feather morphogenesis, local Wnt signaling is required for proper appendage bud formation and local morphogenesis from a flat epithelium; increasing Wnt activation results in randomized and expanded growth zones and ectopic bud formation (*Widelitz et al., 1999*; *Chu et al., 2004*). Similarly, Wnt signaling defines localized growth zones during liver development and regeneration (*Suksaweang et al., 2004*) and expression of a constitutively active β-catenin expands local growth zones and stem cell populations and increases total liver size (*Blanchard et al., 2009*). Our results indicate that *local* Wnt signaling in a tissue experiencing moderate levels of *global* mechanical strain could result in local hotspots of mitosis through increased levels of Wnt/β-catenin signaling.

Levels of strain higher than those applied here have been observed during dorsal closure, germband extension, and neurulation in *Drosophila* and zebrafish (*Blanchard et al., 2009*). Thus, in tissues experiencing high levels of strain (30–50% strain), cell division and tissue expansion may not require Wnt signaling. Indeed, in quiescent MDCK monolayers, 50% uni-axial stretch was sufficient to increase the fraction of cycling cells and number of cell divisions *Streichan et al. (2014)*. Therefore, while high levels of Wnt signaling and mechanical force could independently drive proliferation and tissue expansion, growth-generated mechanical forces during development could also combine with local (Wnt) signaling to refine areas of growth during tissue morphogenesis and during adult tissue repair.

## Materials and methods

### Cell culture and stable cell lines

The cell lines in this study include Madin-Darby canine kidney type II G (MDCK) cells (*Mays et al., 1995*), an MDCK TOPdGFP reporter cell line (*Maher et al., 2009*), and an MDCK FUCCI cell cycle reporter cell line (*Streichan et al., 2014*). No cell lines were used from the International Cell Line

Authentication committee and all cell lines are mycoplasma free. MDCK cells were grown in DMEM with low glucose and 200 µM G418 (TOPdGFP) to maintain stable expression of reporter constructs of interest. MDCK cells stably expressing the FUCCI cell cycle reporter (*Streichan et al., 2014*) were not kept under selection, and were only kept in culture for 8 weeks before thawing an early passage population.

## Preparation of monolayers before mechanical strain

Very dense, quiescent monolayers of MDCK cells were formed using a calcium switch method, as described previously (*Benham-Pyle et al., 2015*). Briefly, MDCK cells were plated on Collagen-I coated flexible PDMS substrates at ~1000 cells/mm in low calcium (5 µM) DMEM with low glucose (200 mg/L). After 60 min, the medium was removed and replaced by DMEM with low glucose and a normal calcium level (1.8 mM), resulting in a super-confluent monolayer in which after 36–48 hr, when mechanical strain was applied, >95% of cells were in G0 (*Benham-Pyle et al., 2015*). Live imaging was performed using the biaxial live cell stretcher, and immunofluorescence images of fixed monolayers was prepared from samples stretched on the integrated strain array (see below, and also [*Benham-Pyle et al., 2015*; *Simmons et al., 2011*]).

## The Integrated strain array (ISA)

We used the ISA in this study so that we could directly compare results to our previous publication (*Benham-Pyle et al., 2015*). The ISA applies a maximum of 15% strain to cell monolayers, and maximum strain was used in all experiments. The ISA is designed for high-throughput analysis of up to 5 different strain levels, each applied simultaneously to 5 wells; the surface area of silicone substrates in each PDMS well is 0.81 cm$^2$ (~equal to the well of a 96-well plate) but the surface area strained is less since the surface area of the pillar used to stretch the substrates is smaller, depending on the amount of strain applied (*Simmons et al., 2011*). Thus the cell population includes a uniform area under strain, surrounded by a variable area, depending on the level of strain, outside the pillar that is not strained. The small surface area, and hence small number of cells, coupled with 2 populations of strained and non-strained cells precludes rigorous biochemical analysis of the entire cell population. To circumvent these limitations, the quantitative methods used to measure protein levels from fluorescence images only obtained data from areas of cells over the pillar and hence under strain, and the custom MATLAB scripts provided an unbiased analysis of <u>all</u> cells in those areas comprising ~500–15,000 cells. Exact cell numbers for each experiment have been detailed in the source data files uploaded to the eLife web-site.

## Design, fabrication, and calibration of a biaxial live imaging cell stretching device

Live imaging of Fucci MDCK monolayers was achieved using a biaxial live cell stretching device (*Figure 1—figure supplement 1*). The silicon bi-axial device was made by mixing PDMS (Sylgard 182; Dow Corning, Inc.) with 10:1 mixing ratio of elastomer and curing agent that was poured over a 3-D printed mold. The mold was degassed for 1 hr and then baked at 65°C overnight. The device from the mold was smoothed by stamping on a thin amount of uncured PDMS and baked at 65°C for 3 hr. The device was then plasma bonded (PDC-32G, Harrick Plasma Inc.) to a prefabricated silicone membrane (SMI Manufacturing Inc., 125 µm thickness). The cell loading well was 8.5 mm in diameter and was separated from the 4 mm pneumatic chamber by a wall 2 mm thick and 3 mm high. Strain was applied to the cell culture membrane by applying vacuum to the pneumatic chamber to deflect the walls outward from the cell loading well and consequently stretched the suspended membrane and attached cell monolayer.

Applied strain using the biaxial live imaging cell stretcher was measured experimentally from displacements of 0.5 µm fluorescent microspheres (F8812, Thermo Fisher Scientific Inc., Waltham, MA) placed on the PDMS surface, or imaging cell nuclei in a Fucci MDCK monolayer (*Figure 1—figure supplement 1C–F*). A pressure regulator (P3RA17132NNKN, Parker Valve, Inc., Cleveland, OH) was used to control pressure in the pneumatic channel and the pressure differential to atmospheric pressure was monitored using a pressure gauge (Ashcroft 25 1009 SW 02L, Ashcroft gauge Inc., Stratford, CT). Maximum strain to the device was measured at 65 kPa of applied pressure. This level of vacuum pressure resulted in 8.5% strain measured by microspheres (*Figure 1—figure*

*supplement 1C, D*) and an average of 8.6% increase in distance to nearest neighbor of living cells in the monolayer (*Figure 1—figure supplement 1F*). Distance to nearest neighbor was calculated for each cell in an image frame, and was equal to the average distance from the nucleus centroid to the centroids of the nearest 7 cell nuclei (See *Figure 1—figure supplement 1E*). The live cell stretcher applies a maximum of 8.5% strain to cell monolayers, and maximum strain was used in all experiments.

## DNA damage, inhibitor, and Wnt3A studies

The DNA damaging agent methyl methanesulfonate (MMS, Sigma-Aldrich) was used as a positive control in DNA damage studies. Cells were treated with 0.1% (v/v) MMS for 2 hr prior to fixation and processing for imaging. The small molecule inhibitor iCRT3 (*Gonsalves et al., 2011*) (SML0211; Sigma-Aldrich, St. Louis, MO) was used to inhibit β-catenin/TCF interactions and their transcriptional activity. iCRT3 (25 µM) and DMSO controls were added to strain array wells 15 min prior to the application of strain. Src Inhibitors PP2 (10 µM; EMD Millipore, Germany) and SU6656 (10 µM, Sigma-Aldrich), CKI inhibitor D4476 (10 µM; Abcam), EGFR inhibitor PD153035 (2.5 µM, Abcam, United Kingdom), and DMSO controls were added to strain device wells 15 min prior to strain application. Fresh inhibitor was added to all experiments every 12 hr. Wnt3A-expressing mouse L cells and parental L cells were grown as indicated (ATCC CRL-2647 and CRL-2648), and conditioned media was obtained as indicated in the comments section of ATCC CRL-2647. Conditioned media from Wnt3A-expressing and parental L cells was added at a 1:2 dilution to MCDK cell culture medium (described above) 15 min prior to strain application.

## Immunofluorescence staining

 MDCK cells, plated on collagen I-coated PDMS, were fixed in 4% (v/v) paraformaldehyde for 15 min, permeabilized in 0.5% (v/v) Triton X-100 for 7 min, and blocked in PBS containing 0.2% (w/v) BSA and 1% goat or donkey serum at room temperature for 60 min. Primary antibodies used for immunofluorescence staining were: β-catenin (610154; BD Biosciences, San Jose, CA), phospho-γ H2A.X (ab11174; Abcam), p53 (92825; Cell Signaling Technology, Danvers, MA), p53BP1 (sc-10915; Santa Cruz Biotechnology, Dallas, TX), and pY654 β-catenin/mouse (Developmental Studies Hybridoma Bank). EdU incorporation was performed using a Click-iT Plus EdU Alexa Fluor 647 Imaging Kit (C10339; Molecular Probes/Invitrogen, Waltham, MA) as directed by the manufacturer.

## Imaging

Immunofluorescence images were acquired with a Zeiss (Jena, Germany) Axiovert 200 inverted microscope equipped with a Mercury lamp and a 63X objective (Olympus, Tokyo, Japan), and acquired with MicroManager Microscopy Software. Fluorescence live-cell images were acquired on a custom-built Zeiss Axiovert 200 M inverted wide field epifluorescence microscope (Intelligent Imaging Innovations (3i), Denver, CO, USA) equipped with a Hamamatsu C11440 Digital camera (Orcaflash4.0OLT), an x, y motorized stage with harmonic drive z-focusing, and a quad filter set for DAPI, FITC, Cy3 and Cy5 laser system (Andor Technology, South Widsor, CT, USA). Live cells were imaged in DMEM without Phenol Red at 37°C with a 20X air objective for tracking cellular movement; images were acquired using Slidebook (3i) software.

## Image quantification

The chosen cell cycle markers, the transcriptional reporter, and (transcriptionally active) β-catenin localized in the cell nucleus. Therefore, the strategy for image quantification was nuclear segmentation (using Hoechst staining) followed by pixel intensity calculation for each nucleus; this method excluded cytoplasmic and plasma membrane staining. The nuclear pixel intensity of all biomarkers (β-catenin, TBSmCherry, TOPdGFP, EdU, and Ki67) was analyzed on a single cell basis using a previously described image-processing routine in MATLAB (*Benham-Pyle et al., 2015*). The resulting data were exported from MATLAB to a text file for plotting and further analysis. Exported data for each nucleus included the object identifier, object area, and pixel intensity of up to 3 analyzed imaging channels. Cytoplasmic staining of pY654 β-catenin was also quantified using a custom image-processing routine in MATLAB. Briefly, nuclei were identified and segmented using images of Hoechst stained cells. Pixels associated with nuclear objects were eliminated from analysis and

remaining pixels were quantified to produce a whole image cytoplasmic pixel intensity that was exported from MATLAB to a text file for plotting and further analysis. Single Cell Tracking of Geminin-positive cells in live imaging movies (*Figures 1* and *4–7*) was performed using the TrackMate plugin (FIJI). Exported data were processed using MATLAB to report the timing of an object's appearance and disappearance following the start of a movie and the time spent by the cell in S/G2.

## Statistical analysis

Image quantification for signaling responses was represented as percent cells positive for the relevant biomarker and the threshold intensity separating positive and negative cells was kept constant between compared conditions. Statistics compared the mean percent of positive cells between multiple independent experiments using an unpaired t-test and the Holm-Sidak method to correct for multiple comparisons. When reporting # of occurrences of cell cycle progression, statistics compared the number of occurrences per 0.43 mm$^2$ (movie frame) over 24 hr between conditions using an unpaired t-test. In cases where the reported metric was an absolute distance to a nearest neighbor or a pixel intensity value relative to a no strain control, data were not assumed to be normally distributed and a Kolmogorov-Smirnoff (KS) test was used to compare distributions. All experiments on the ISA were repeated at least 3 times, and the data from all experiments used in the statistical analysis and figure presentations. For Fucci MDCK monolayers, at least 6 replicate movies were taken from at least 2 independent experiments and the data from all experiments used in the statistical analysis and figure presentations.

## Acknowledgements

This work was supported by the National Science Foundation (NSF) under grant EFRI MIKS to BLP and WJN (1136790), a NSF Pre-doctoral Fellowship (DGE-114747) and a Stanford Lieberman Fellowship to BWB-P, a NIH Training Grant (T32GM007276) to KCH, Stanford Bio-X SIGF Fellowships to KCH and JYS, and a grant from the National Institute of Health to WJN (NIH 11R35GM118064-01).

## Additional information

### Competing interests

WJN: Reviewing editor, *eLife*. The other authors declare that no competing interests exist.

### Funding

| Funder | Grant reference number | Author |
| --- | --- | --- |
| National Science Foundation | 1136790 | Beth L Pruitt<br>William James Nelson |
| National Science Foundation | Graduate Student Fellowship | Blair W Benham-Pyle |
| National Institutes of Health | T32GM007276 | Kevin C Hart |
| National Institutes of Health | 11R35GM118064-01 | William James Nelson |
| Stanford University | Bio-X Graduate Fellowship | Joo Yong Sim<br>Kevin C Hart |
| Stanford University | Lieberman Graduate Fellowship | Blair W Benham-Pyle |
| National Science Foundation | DGE-114747 | William James Nelson |

The funders had no role in study design, data collection and interpretation, or the decision to submit the work for publication.

### Author contributions

BWB-P, Conception and design, Acquisition of data, Analysis and interpretation of data, Drafting or revising the article; JYS, Design of Biaxial Live Cell Stretcher, Drafting or revising the article; KCH, Fluorescent Bead Calibration of Biaxial Live Cell Stretcher, Drafting or revising the article; BLP,

Analysis and interpretation of data, Drafting or revising the article; WJN, Conception and design, Drafting or revising the article

## Author ORCIDs

Beth L Pruitt, http://orcid.org/0000-0002-4861-2124
William James Nelson, http://orcid.org/0000-0003-3039-3776

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
