## [Decision Letter]

Thank you for submitting your article "Increasing β-catenin/Wnt3A Activity Levels Drive Mechanical Strain-Induced Cell Cycle Progression through Mitosis" for consideration by *eLife*. Your article has been reviewed by three peer reviewers, one of whom, Reinhard Fässler (Reviewer #1), is a member of our Board of Reviewing Editors and the evaluation has been overseen by Vivek Malhotra as the Senior Editor.

The reviewers have discussed the reviews with one another and the Reviewing Editor has drafted this decision to help you prepare a revised submission.

Summary:

Benham-Pyle and co-workers reported in an earlier paper that forces applied to a monolayer of MDCK epithelial cells induced rapid cell cycle entry and progression to S through the transcriptional activity of first Yap1 and then β-catenin. In the present paper, the authors extend their observations and show that force-induced Src activity and an additional elevation of β-catenin, achieved either by treating the monolayer with Wnt3a or a Casein Kinase I inhibitor (both of which stabilises β-catenin and increases nuclear β-catenin levels), are required to enable mitotic entry and successful cell division.

Essential revisions:

1) Confirm b-catenin phosphorylation and degradation with quantitative Western blots,

2) Confirm that DNA damage plays no role in G2/S arrest with quantitative Western blots for active p53 and ATM/ATR,

3) Confirm that indeed Src and not EGFR (PP2 inhibits both functions) is activated upon strain,

4) Investigate whether increased strain can bypass the need for high, local Wnt signalling. If it can, test whether G2/S arrest is induced by inhibiting Src or enhanced by inhibiting CKI,

5) Better discuss the relevance of this in vitro crosstalk of Wnt and force for cell division in vivo.

[Editors' note: further revisions were requested prior to acceptance, as described below.]

Thank you for resubmitting your work entitled "Increasing β-catenin/Wnt3A Activity Levels Drive Mechanical Strain-Induced Cell Cycle Progression through Mitosis" for further consideration at *eLife*. Your revised article has been favorably evaluated by Vivek Malhotra (Senior editor) and three reviewers, one of whom, Reinhard Fässler, is a member of our Board of Reviewing Editors.

The manuscript has been improved but there are some remaining issues that need to be addressed before acceptance, as outlined below:

Specifically, the reviewers were disappointed that you did not test whether higher strain allows cells to proceed into mitosis in the absence of Wnt signalling, and whether the G2/S arrest is induced by inhibiting Src or enhanced by inhibiting CKI under higher strains. We appreciate that it is technically not trivial to apply the higher strain requested. Therefore, we ask to revise the text of the manuscript and discuss in extenso that increasing the strain above the level used in the current study may well occur in vivo (Blanchard et al. 2009, Nature Methods) and that in such situations the role of Wnt may be less important or even fully dispensable for cells to pass through S/G2 and mitosis. In this context, the Streichan et al. citation serves as an excellent reference for the potential impact of higher strain the requirement of Wnt.

---

## [Author Response]

[…]

Essential revisions:

1) Confirm b-catenin phosphorylation and degradation with quantitative Western blots,

We were unable to perform biochemical studies with the integrated strain array (ISA) used in the current studies, and our previous study (Benham-Pyle, Pruitt and Nelson, 2015).

We used the ISA in this study so that we could directly compare results to our previous publication (Benham-Pyle, Pruitt and Nelson, 2015). The ISA is designed for high-throughput analysis of up to 5 different strain levels, each applied simultaneously to 5 wells; the surface area of silicone substrates in each PDMS well is 0.81 cm^2^ (~ equal to the well of a 96-well plate) but the surface area strained is less since the surface area of the pillar used to stretch the substrates is smaller, depending on the amount of strain applied (Simmons et al., 2011). Thus the cell population includes a uniform area under strain, surrounded by a variable area, depending on the level of strain, outside the pillar that is not strained. The small surface area, and hence small number of cells, coupled with 2 populations of strained and non-strained cells precludes rigorous biochemical analysis of the entire cell population. To circumvent these limitations, the quantitative methods used to measure protein levels from fluorescence images only obtained data from areas of cells over the pillar and hence under strain, and the custom MATLAB scripts provided an unbiased analysis of ALL cells in those areas comprising ~500 – 15000 cells. Exact cell numbers for each experiment have been included in the Source Data files for Figures and uploaded to the *eLife* web-site.

In addition, the highest level of strain that our devices can apply (~10%) resulted in 20-30% of cells activating β-catenin signaling and re-entering the cell cycle. Therefore, even if the strained populations could be isolated and pooled, data from a western blot of the whole cell population would be compromised by the heterogeneity in cell responses to strain, whereas the immunofluorescence methods and quantitation with MATLAB scripts provided the single cell resolution and sensitivity required to quantify the phenotype.

To address these concerns, we have included statements in Results and Materials and methods sections to clarify the technical limitations of the ISA and its incompatibility with biochemical analyses. For added text, see Results section, “The small surface area (0.81cm2) of the ISA does not provide sufficient cell numbers for biochemical characterization…”. We also added the text above in the Methods and Materials section “We used the ISA in this study so…..”.

2) Confirm that DNA damage plays no role in G2/S arrest with quantitative Western blots for active p53 and ATM/ATR,

For the reasons stated in response to point #1, we could not add quantitative western blots to our revision.

However, as suggested by the reviewers, we added additional markers for DNA damage – p53 and p53 Binding Protein 1 (p53BP1). In MDCK cells, activation of the DNA Damage Response results in increased accumulation of both p53 and p53BP1. While both p53 and p53BP1 levels increased significantly in cells treated with the DNA damaging agent MMS, no significant increases in levels of either p53 or p53BP1 were observed in strained monolayers (Figure 2). These results confirmed our conclusion from our original analysis of phospho-γH2A.X that DNA damage does not play a role in strain-induced G2/S arrest. These data have been added to Figure 2, and discussed in Results and Discussion.

For added text, see Results section, “The monolayers were stained for the DNA damage associated histone variant phospho-γH2A.X (40,41) and the DNA repair proteins p53 and p53 binding protein 1 (Zhang et al. 2009; Wagstaff et al. 2016) … “, and in the Discussion section, “…..level of DNA damage in those cells, measured by levels of p53, p53BP1 and phospho-γH2A.X, was insignificant compared….”.

3) Confirm that indeed Src and not EGFR (PP2 inhibits both functions) is activated upon strain,

The experiments presented in the text with PP2 inhibition were repeated with the more specific Src Inhibitor SU6656 (Figure 3 and Figure 3—figure supplement 1), and the EGFR inhibitor PD153035 (Figure 3—figure supplement 4 and Figure 3—figure supplement 5).

Results using the more specific Src inhibitor SU6656 were similar to those observed with PP2 treatment. SU6656 treatment blocked strain-induced increases in: 1) pY654 β-catenin; 2) β-catenin transcriptional activity; and 3) cell cycle re-entry (EdU incorporation). We replaced the PP2 data previously presented in Figure 3 with the results with SU6656, and included the PP2 results as Supplements to the updated Figure 3. We also added text to describe the data with SU6656 in the Results section, “Mechanical strain of monolayers in the absence of SU6656… Similar results to those with SU6656 treatment were obtained upon treatment with the tyrosine kinase inhibitor PP2…”.

EGFR inhibition with PD153035 treatment reduced by 30-40%, but did not completely block (unlike SU6656), the strain-induced increase in pY654 β-catenin. However, PD153035 treatment had no effect on strain-induced β-catenin transcriptional activity or cell cycle re-entry. See Results section, “To test whether EGFR activation was involved in strain-induced accumulation of pY654 β-catenin and β-catenin signaling, monolayers were treated with the EGFR inhibitor PD153035…”

4) Investigate whether increased strain can bypass the need for high, local Wnt signalling. If it can, test whether G2/S arrest is induced by inhibiting Src or enhanced by inhibiting CKI,

The ISA and live strain device used here are capable of applying a maximum of ~15% and ~10% strain, respectively, and these maximum strain levels were used in our studies. These levels of strain are recognized as within the physiological level in vivo. We note that ~ 100% uni-axial strain has been applied previously to MDCK monolayers, and cells strained at that level did proceed through G2 into mitosis (Streichen et al. 2014).

*5) Better discuss the relevance of this* in vitro *crosstalk of Wnt and force for cell division* in vivo.

In the original submission, we were reluctant to speculate too much about the in vivo relevance of our work. Nevertheless, in response to the reviewer’s urging, we expanded the discussion to include more in vivo relevance, and the potential importance of our observations. “It has been a long-standing hypothesis that growth-induced local mechanical cues could influence morphogenesis by modulating growth rates, the direction of growth, or differentiation…”

[Editors' note: further revisions were requested prior to acceptance, as described below.]

[…]

*Specifically, the reviewers were disappointed that you did not test whether higher strain allows cells to proceed into mitosis in the absence of Wnt signalling, and whether the G2/S arrest is induced by inhibiting Src or enhanced by inhibiting CKI under higher strains. We appreciate that it is technically not trivial to apply the higher strain requested. Therefore, we ask to revise the text of the manuscript and discuss in extenso that increasing the strain above the level used in the current study may well occur* in vivo *(Blanchard et al. 2009, Nature Methods) and that in such situations the role of Wnt may be less important or even fully dispensable for cells to pass through S/G2 and mitosis. In this context, the Streichan et al. citation serves as an excellent reference for the potential impact of higher strain the requirement of Wnt.*

We are happy to address this in the revised text in 2 ways:

1). The cell stretchers that we use can apply a maximum of 8% (live imaging) and 15% (ISA) strain. Therefore we cannot apply strains of >30% as requested by the reviewers. We have added the following text to state this technical limitation:

Results section: “The live cell stretcher and ISA were able to apply maximum strains of 8.5% and 15%, respectively. The maximum level of static biaxial stretch was applied and held for up to 24 hours,….”.

Materials and methods section): “The ISA applies a maximum of 15% strain to cell monolayers, and maximum strain was used in all experiments.”

Materials and methods section): “The live cell stretcher applies a maximum of 8.5% strain to cell monolayers, and maximum strain was used in all experiments.”

2). The reviewers asked us “to revise the text of the manuscript and discuss in extenso that increasing the strain above the level used in the current study may well occur in vivo (Blanchard et al. 2009, Nature Methods) and that in such situations the role of Wnt may be less important or even fully dispensable for cells to pass through S/G2 and mitosis.” We have added the following text, with the suggested references, to discuss the possibility that higher levels of strain could induce mitosis in the absence of Wnt signaling:

Discussion section: “Our results indicate that local Wnt signaling in a tissue experiencing moderate levels of global mechanical strain could result in local hotspots of mitosis through increased levels of Wnt/β-catenin signaling.

Levels of strain higher than those applied here have been observed during dorsal closure, germband extension, and neurulation in *Drosophila* and zebrafish (Blanchard et al. 2009). Thus, in tissues experiencing high levels of strain (30-50% strain), cell division and tissue expansion may not require Wnt signaling. Indeed, in quiescent MDCK monolayers, 50% uni-axial stretch was sufficient to increase the fraction of cycling cells and number of cell divisions (Streichan et al. 2014). Therefore, while high levels of Wnt signaling and mechanical force could independently drive proliferation and tissue expansion, growth-generated mechanical forces during development could also combine with local (Wnt) signaling to refine areas of growth during tissue morphogenesis and during adult tissue repair.”